# Tumor-intrinsic response to IFNγ shapes the tumor microenvironment and anti–PD-1 response in NSCLC

Bonnie L Bullock[1], Abigail K Kimball[2], Joanna M Poczobutt[1], Alexander J Neuwelt[1], Howard Y Li[1,3], Amber M Johnson[1], Jeff W Kwak[1], Emily K Kleczko[1], Rachael E Kaspar[2], Emily K Wagner[1], Katharina Hopp[1], Erin L Schenk[1], Mary CM Weiser-Evans[1], Eric T Clambey[2], Raphael A Nemenoff[1]

**Targeting PD-1/PD-L1 is only effective in ~20% of lung cancer patients, but determinants of this response are poorly defined. We previously observed differential responses of two murine K-Ras–mutant lung cancer cell lines to anti–PD-1 therapy: CMT167 tumors were eliminated, whereas Lewis Lung Carcinoma (LLC) tumors were resistant. The goal of this study was to define mechanism(s) mediating this difference. RNA sequencing analysis of cancer cells recovered from lung tumors revealed that CMT167 cells induced an IFNγ signature that was blunted in LLC cells. Silencing *Ifngr1* in CMT167 resulted in tumors resistant to IFNγ and anti–PD-1 therapy. Conversely, LLC cells had high basal expression of SOCS1, an inhibitor of IFNγ. Silencing *Socs1* increased response to IFNγ in vitro and sensitized tumors to anti–PD-1. This was associated with a reshaped tumor microenvironment, characterized by enhanced T cell infiltration and enrichment of PD-L1[hi] myeloid cells. These studies demonstrate that targeted enhancement of tumor-intrinsic IFNγ signaling can induce a cascade of changes associated with increased therapeutic vulnerability.**

## Introduction

The development of immune checkpoint inhibitors has shown great promise in a wide variety of malignancies, including lung cancer. However, only ~20% of unselected non-small cell lung cancer patients respond to monotherapy targeting the Programmed Cell Death Protein 1 (PD-1)/Programmed Death Ligand 1 (PD-L1) axis (Borghaei et al, 2015; Brahmer et al, 2015; Garon et al, 2015). Previous studies have correlated multiple factors with patient response to immunotherapy. These include tumor mutational burden, the presence of neoantigens, PD-L1 expression on the surface of tumor cells and/or surrounding stromal cells, tumor-infiltrating immune cells, and patient smoking status (Ji et al, 2012; Ngiow et al, 2015; Danilova et al, 2016; Gainor et al, 2016; Spranger et al, 2016; Ayers et al, 2017; Corrales et al, 2017).

Importantly, Ayers et al (2017) defined an IFNγ gene signature generated from melanoma patient tumors that correlated with enhanced response to pembrolizumab across multiple cancer types. Although many clinical trials involving single-agent immunotherapy or combination therapies are being performed in non-small cell lung cancer, a mechanistic understanding of determinants of response to these agents is still incomplete. These studies require preclinical models that accurately recapitulate features of human lung cancer.

Our laboratory has used an orthotopic and immunocompetent mouse model to study how K-Ras–mutant lung cancers respond to the immune system (Poczobutt et al, 2016a, 2016b; Li et al, 2017; Kwak et al, 2018). In this model, lung cancer cells derived from C57BL/6J mice are implanted directly into the lungs of syngeneic mice. These cells form a primary tumor after 2–4 wk that metastasizes to the other lung lobes, liver, brain, and mediastinum (Weiser-Evans et al, 2009). This model has the advantage that tumors develop in the appropriate tumor microenvironment (TME) and imitate late-stage disease when most patients are placed on immunotherapy. In addition, the non-synonymous mutational burden in these tumors is comparable with human lung tumors, and significantly higher than genetically engineered mouse models (McFadden et al, 2016), allowing for recognition by the adaptive immune system. We have previously demonstrated differential sensitivity of K-Ras–mutant tumors to anti–PD-1/anti–PD-L1 therapy, with CMT167 tumors showing a strong inhibition and Lewis lung carcinoma (LLC) tumors being generally unresponsive (Li et al, 2017). The responsiveness of these tumors was also dependent on the local TME. CMT167 tumors implanted subcutaneously were resistant to anti–PD-1 therapy, whereas tumors in the lung were eliminated (Li et al, 2017). Thus, this model allows us to define specific mechanisms that determine the response to immunotherapy. In this study, we have focused on how cancer cell–intrinsic response to IFNγ affects the TME and response to anti–PD-1 therapy.

IFNγ is made predominantly by NK cells, type 1 innate lymphoid cells (ILC1), and T cells (Schroder et al, 2004; Cheon et al, 2011). Since the 1990s, it has been shown that IFNγ increases the immunogenicity of some tumors (Cheon et al, 2014). IFNγ binds to cell surface receptors (IFNGR1/IFNGR2) on cancer cells resulting in activation of

[1]Department of Medicine, University of Colorado Anschutz Medical Campus, Aurora, CO, USA    [2]Department of Anesthesiology, University of Colorado Anschutz Medical Campus, Aurora, CO, USA    [3]Veterans Affairs Medical Center, Denver, CO, USA

Correspondence: raphael.nemenoff@ucdenver.edu

JAK1 and JAK2 and phosphorylation of STAT1 (Cheon et al, 2014). Activated STAT1 dimers translocate to the nucleus to initiate waves of transcription that can lead to enhanced MHC class I and II presentation on tumor cells and increased chemokine expression. Global loss of IFNγ is detrimental to tumor surveillance in mice, as $Ifn\gamma^{-/-}$ mice develop tumors more quickly than their $Ifn\gamma^{+/+}$ counterparts in the setting of carcinogen-induced or spontaneously arising tumors (Kaplan et al, 1998; Shankaran et al, 2001). Tumors that are insensitive to IFNγ can grow equally well in $Ifn\gamma^{-/-}$ or $Ifn\gamma^{+/+}$ mice, suggesting that host response does not completely alter the growth of these tumors. Thus, it has been speculated that many tumors develop mutations in the IFNγ signaling pathway to evade the immune system. Recent studies have shown that ~30% of both melanoma and lung carcinomas have at least one mutation in the IFNγ pathway, including JAK1, IFNGR1, or IFNGR2 (Cheon et al, 2014), and resistance to checkpoint inhibitors in patients is associated with JAK1/2 mutations (Shin et al, 2017).

We hypothesized that intrinsic differences in the responsiveness of cancer cells to IFNγ, distinct from other features of these cells, define the nature of the TME and control sensitivity of lung tumors to immunotherapy. In this study, we demonstrated that by altering responsiveness of murine lung cancer cells to IFNγ, we could define changes in the TME that regulate responsiveness to anti–PD-1 therapy.

## Results

### LLC cells exhibit a blunted response to IFNγ in vitro and in vivo compared with CMT167

We hypothesized that the differential response of CMT167 versus LLC orthotopic lung tumors to anti–PD-1 therapy was mediated at least in part through inherent differences in the cancer cells and how they respond to signals coming from the TME. To define these changes, we recovered cancer cells from orthotopically implanted tumors and compared their transcriptional profile with identical cancer cells grown in vitro. CMT167 or LLC cells were injected into the lungs of transgenic GFP-expressing C57BL/6J mice. After tumors were established, the GFP-negative cancer cell population was recovered by FACS of single-cell suspensions made from tumor-bearing lungs. RNA isolated from recovered cells and from identical cells grown in vitro was analyzed by RNA sequencing (RNA-Seq). After gene set enrichment analysis, we determined that the IFNγ signaling pathway was up-regulated in CMT167 cancer cells relative to LLC cells in vivo (Fig 1A and Table S1), suggesting that these cells have a differential response to IFNγ. We also confirmed that IFNγ was present in LLC and CMT167 tumor homogenate (Fig S1A). Examination of RNA-Seq data revealed no detectable mutations in the IFN signaling pathways in either cell line (Table S2). Both cell lines also expressed IFNγ receptors and JAK/STAT machinery, implicating other potential alterations in intracellular signaling.

To validate our RNA-Seq data, we compared the responsiveness of these two cell lines to IFNγ treatment in vitro. CMT167 cells showed a more robust and sustained induction of phospho-STAT1

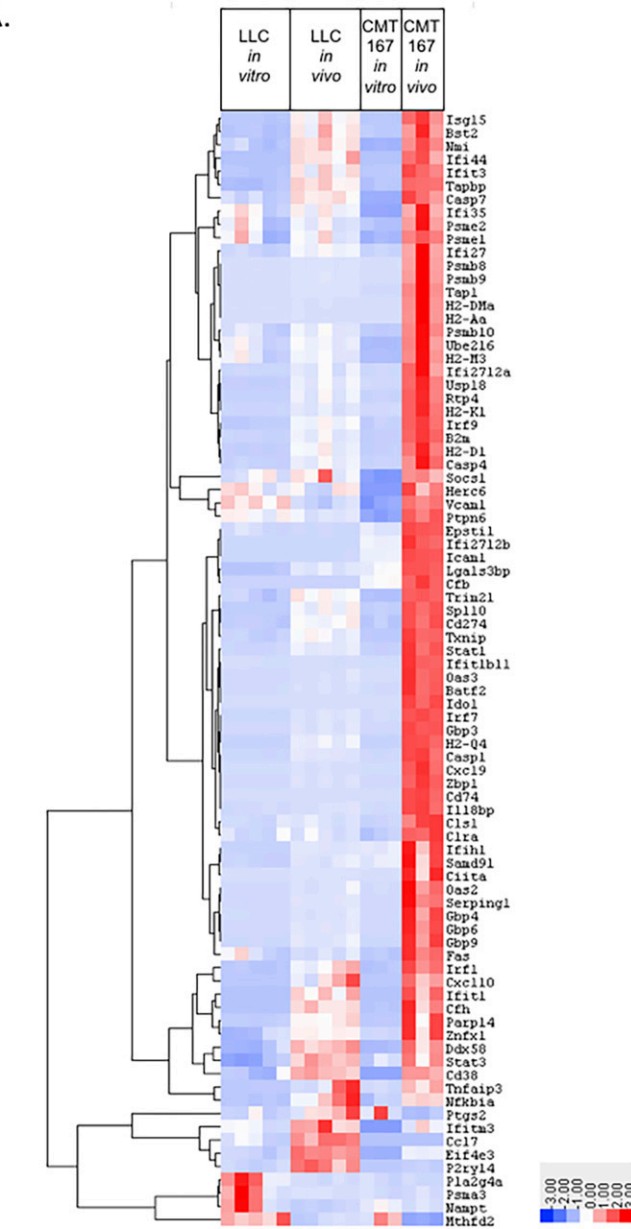

**Figure 1.  LLC cells exhibit a blunted response to IFNγ in vitro and in vivo compared with CMT167.**
CMT167 or LLC cells were orthotopically injected into the left lung lobe of transgenic GFP-expressing C57BL/6J mice and were grown for either 2 (LLC) or 3 wk (CMT167). Tumor-bearing lung lobes were isolated and made into single-cell suspensions containing both GFP-positive (host cells) and GFP-negative (cancer cells). First, RNA was isolated from identical cancer cells grown in passage (in vitro condition). Second, RNA was isolated from recovered GFP-negative cancer cells (isolated via FACS-in vivo condition). Third, RNA was run for RNA-Seq from both conditions. The CMT167 condition had three experimental replicates per in vitro and in vivo conditions with five tumor-bearing lung lobes pooled per *in vivo* experimental replicate (15 mice used total). The LLC condition had five experimental replicates per in vitro and in vivo conditions with four tumor-bearing lung lobes pooled per in vivo experimental replicate (20 mice used total). **(A)** The "HM_IFNg Response" pathway containing 80 genes from the Broad Institute Molecular Signatures Database was used to generate a heat map of differentially expressed genes between the LLC and CMT167 in vitro and in vivo experimental conditions.
Source data are available for this figure.

(p-STAT1) levels upon treatment with IFNγ compared with LLC cells (Fig 2A). By message, CMT167 also showed greater induction of four downstream IFNγ response genes (*Cxcl9, Cxcl10, Cd274,* and *Ciita*) compared with LLC cells (Fig 2B–E). By protein, CMT167 cells induced CXCL9, CXCL10, PD-L1, and MHC class II (a read-out of CIITA activity) to a significant degree over LLC cells, whereas both cell lines induced two MHC class I genes (H2-D and H2-K) (Fig S1B–H). Importantly, upon treatment with ruxolitinib, a JAK1/2 inhibitor, protein expression of several genes was abrogated, indicating that their expression was predominately JAK-STAT dependent (Fig S1B–H) in both cell lines. Collectively, these data suggest that responsiveness to IFNγ signaling is associated with sensitivity to anti–PD-1 therapy in our model.

### Silencing *Ifngr1* in CMT167 confers decreased response to IFNγ and resistance to anti–PD-1 therapy

The induction of an IFNγ signature in CMT167 in vivo suggests that these cancer cells respond to and induce IFN-dependent effectors. Because LLC and CMT167 cells both express *Ifngr1* and *Ifngr2* via RNA-Seq (Fig S2A and B), we confirmed their expression by immunoblot (Fig S2C). Next, to determine how responsiveness to IFNγ affects tumor growth and response to checkpoint inhibitors, *Ifngr1* was silenced in CMT167 cells using two separate shRNAs against murine *Ifngr1* and a nontargeting control vector. Expression of *Ifngr1* was decreased by ~80% with both shRNA constructs (CMT-sh68sc3 and CMT-sh69sc2) compared with the nontargeting control (CMT-NT) cell line (Figs 3A and S2D). Importantly, both shRNA knockdowns had decreased induction of p-STAT1 in response to IFNγ stimulation (Fig 3B) and decreased induction of downstream IFN response genes (*Cxcl9, Cxcl10, Cd274, Ciita,* and *Socs1*) (Figs 3C–F and S2E–I).

We selected one knockdown, CMT-sh68sc3, for in vivo studies and validated that PD-L1 protein expression was decreased in these cells in vitro (Fig S2J). Equal numbers of CMT-sh68sc3 or CMT-NT cells were implanted into the lungs of syngeneic WT mice, were then treated with either a control IgG2a antibody or an antibody targeting PD-1, starting 7 d post tumor cell injection. After 4 wk, we found that CMT-NT tumors treated with anti–PD-1 were almost completely eliminated similar to the published CMT167 parental line (Fig 3G) (Li et al, 2017). However, treatment of CMT-sh68sc3 *Ifngr1* KD tumors with anti–PD-1 had no significant effect on tumor size (Fig 3G). We previously reported anti–PD-1 treatment of CMT167 tumors results in nests of infiltrating T cells associated with tumor elimination (Li et al, 2017). Although we observed similar patterns of T cell infiltration in CMT-NT tumors with anti–PD-1 treatment, this was not observed in the CMT-sh68sc3 tumors (Fig S2K and L). These data indicate that the IFNγ responsiveness of CMT167 cells is critical for their response to immunotherapy, although we cannot completely rule out off-target shRNA effects.

### Silencing *Socs1* in the LLC line confers increased response to IFNγ in vitro

Because the lack of an IFNγ response in LLC cells is not due to lack of receptor expression (Fig S2A and B), we examined differences in expression of putative regulators of the IFNγ pathway between the

responsive CMT167 and unresponsive LLC cell lines. We determined that at baseline, LLC cells expressed markedly higher levels of *Socs1,* or suppressor of cytokine signaling 1, which is a critical negative regulator of interferon signaling via RNA-Seq (Fig 4A) (Beaurivage et al, 2016; Liau et al, 2018). We confirmed that LLC cells expressed higher levels of SOCS1 protein relative to CMT167 cells in vitro under control conditions or after stimulation with IFNγ (Fig 4B). These data led us to hypothesize that high baseline levels of SOCS1 mediate at least in part the unresponsiveness of LLC cells to IFNγ. Thus, silencing *Socs1* should increase LLC cells' response to IFNγ in vitro and potentially alter their response to checkpoint inhibitors in vivo.

*Socs1* expression in LLC cells was silenced using two separate shRNAs and a nontargeting control construct (LLC-sh20, LLC-sh21, and LLC-NT). As anticipated, both knockdowns exhibited enhanced STAT1 signaling at early and late time points after IFNγ treatment as determined by expression of p-STAT1 compared with LLC-NT cells (Figs 4C and S3A). In addition, knockdown variants had decreased *Socs1* mRNA and protein expression (Figs 4D and E, and S3B), whereas levels of *Ifngr1* were unchanged (Fig S3C). Compared with LLC-NT cells, we found that induction of multiple downstream IFNγ response genes (*Cxcl9, Cxcl10, Cd274, and Ciita)* was enhanced in both knockdowns (Fig S3D–G). We chose LLC-sh21 cells for further studies and validated that upon IFNγ stimulation, they had increased protein expression of CXCL9, CXCL10, PD-L1, MHC Class II, H2-D, and H2-K in vitro relative to LLC-NT cells. Induction of these genes by IFNγ was completely inhibited by Ruxolitinib in the LLC-sh21 cells, indicating JAK/STAT–dependent mechanisms (Fig S4A–F). These data collectively indicate that LLC cells are refractory to IFNγ signaling because of high basal levels of SOCS1. In addition, *Socs1* knockdown in LLC cells sensitizes them to IFNγ by increasing the magnitude and duration of JAK/STAT signaling.

### *Socs1* KD tumors show enhanced response to anti–PD-1 therapy

To determine if altering the sensitivity of LLC cells to IFNγ affects tumor growth in vivo and responsiveness to anti–PD-1, we implanted equal numbers of LLC-NT or LLC-sh21 cells into the lungs of syngeneic WT mice. Tumors were allowed to establish for 1 wk and were then treated with either an anti–PD-1 antibody or an isotype control antibody (IgG2a) for 2 wk (as above). Similar to the parental LLC line as previously published (Li et al, 2017), there was no significant difference in primary tumor volume between the LLC-NT tumors treated with anti–PD-1 or isotype control after 3 wk (Fig 4F). However, in mice harboring LLC-sh21 tumors, treatment with the anti–PD-1 antibody decreased primary tumor volume by more than 80%, a statistically significant difference compared with all the other experimental groups (Fig 4F). To determine if these effects were specific to the lung TME, we analyzed the response of LLC-sh21 cells implanted subcutaneously to anti–PD-1 therapy. Unlike what was observed in orthotopic lung tumors, subcutaneous LLC-sh21 tumors were resistant to anti–PD-1 therapy (Fig 4G). This is similar to our previous data showing that the sensitivity of CMT167 tumors to anti–PD-1 therapy was specific to tumors implanted into the lung, whereas identical cells implanted subcutaneously were resistant (Li et al, 2017). These data suggest enhanced responsiveness of LLC tumors to anti–PD-1 is dependent on critical features of the lung TME that are absent in subcutaneous models.

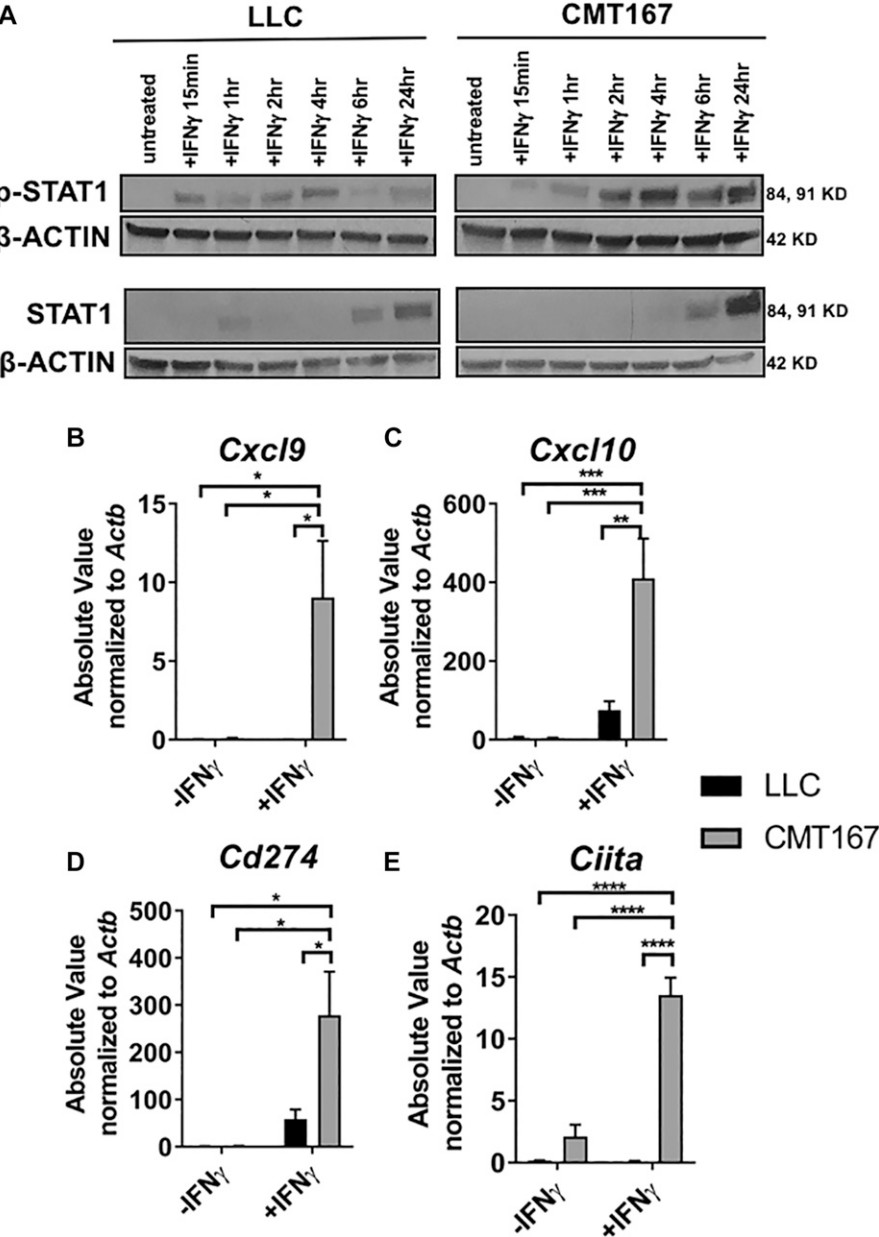

Figure 2. In vitro treatment with IFNγ recapitulates RNA-Seq differences in the IFNγ signaling pathway in LLC and CMT167 cells.
**(A)** Immunoblots of LLC or CMT167 cells treated with ±10 ng/mL IFNγ in vitro for a time course ranging from 15 min to 24 h, showing p-STAT1 and total STAT1 expression compared with the housekeeping gene β-ACTIN. LLC or CMT167 cells were treated with ±100 ng/ml IFNγ in vitro for 24 h followed by isolation of RNA and qRT-PCR. **(B–E)** mRNA levels of (B) *Cxcl9*, (C) *Cxcl10*, (D) *Cd274*, and (E) *Ciita* are shown as absolute values (SQ values) normalized to the housekeeping gene *Actb*. Statistics compare all groups ± IFNγ. Error bars represent the mean of the data ± SEM after a two-way ANOVA (B–E) (*$P < 0.05$, **$P < 0.01$, ***$P < 0.001$, and ****$P < 0.0001$).
Source data are available for this figure.

## *Socs1* KD tumors show alterations in multiple populations

To define changes in the TME, which are dependent on the IFNγ-responsiveness of the LLC cells, we used mass cytometry (CyTOF) to characterize both CD45+ and CD45− populations from LLC-NT and LLC-sh21 tumors before treatment with anti–PD-1 using a panel of 39 different antibodies. Three independent isolations using three mice/isolation were analyzed. There were no significant differences in tumor size between LLC-NT and LLC-sh21 at this time point (Fig S5A). Using the PhenoGraph algorithm, which allows unbiased clustering of events based on cellular distribution and phenotype, we identified 35 distinct clusters in the naïve, LLC-NT, and LLC-sh21 experimental conditions (Table S3) (Levine et al, 2015). Fig 5A depicts the tSNE plot for all samples, with clusters colored by phenotype.

Phenotypes were defined based on the expression level of cellular markers (parameters) (Figs S5B and S6). No significant differences were noted between experimental replicates (Fig S5C).

Fig 5B shows these data as percentages in pie graph format. Because cancer cells do not express a unique cell surface marker, we defined them as CD45− cells that were absent in samples from naïve mice and highly enriched in the LLC-NT and LLC-sh21 tumor-containing samples (Fig 5C). Further examination of cancer cell clusters revealed two cancer clusters present in both LLC-NT and LLC-sh21 tumors that were defined by differential Ki67 expression (Cluster #5, Ki67+; Cluster #35, Ki67−) (Fig S5D). Interestingly, we observed a reduction in the percentage of events that were cancer cells in LLC-sh21 compared with LLC-NT (17.54% LLC-NT to 11.07% LLC-sh21) replicates as well as a decreased frequency of Ki67+

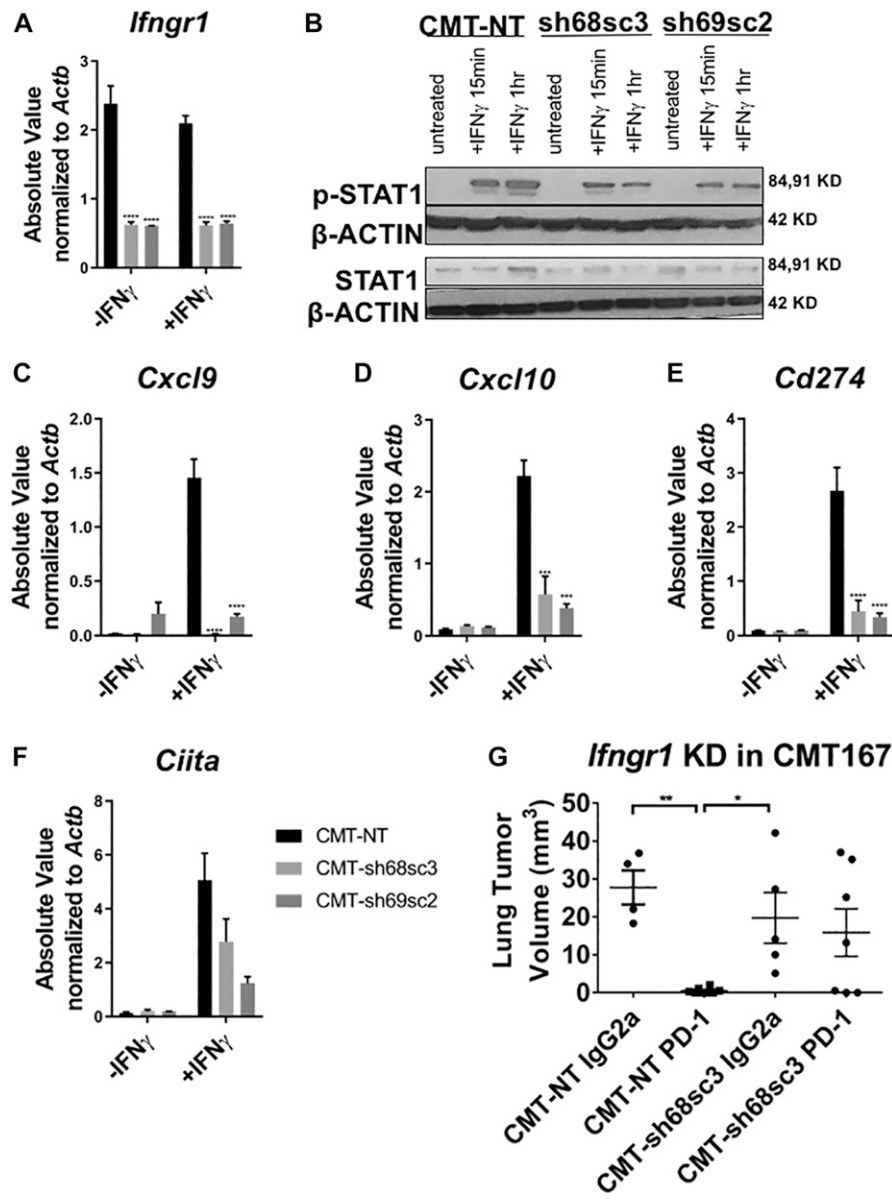

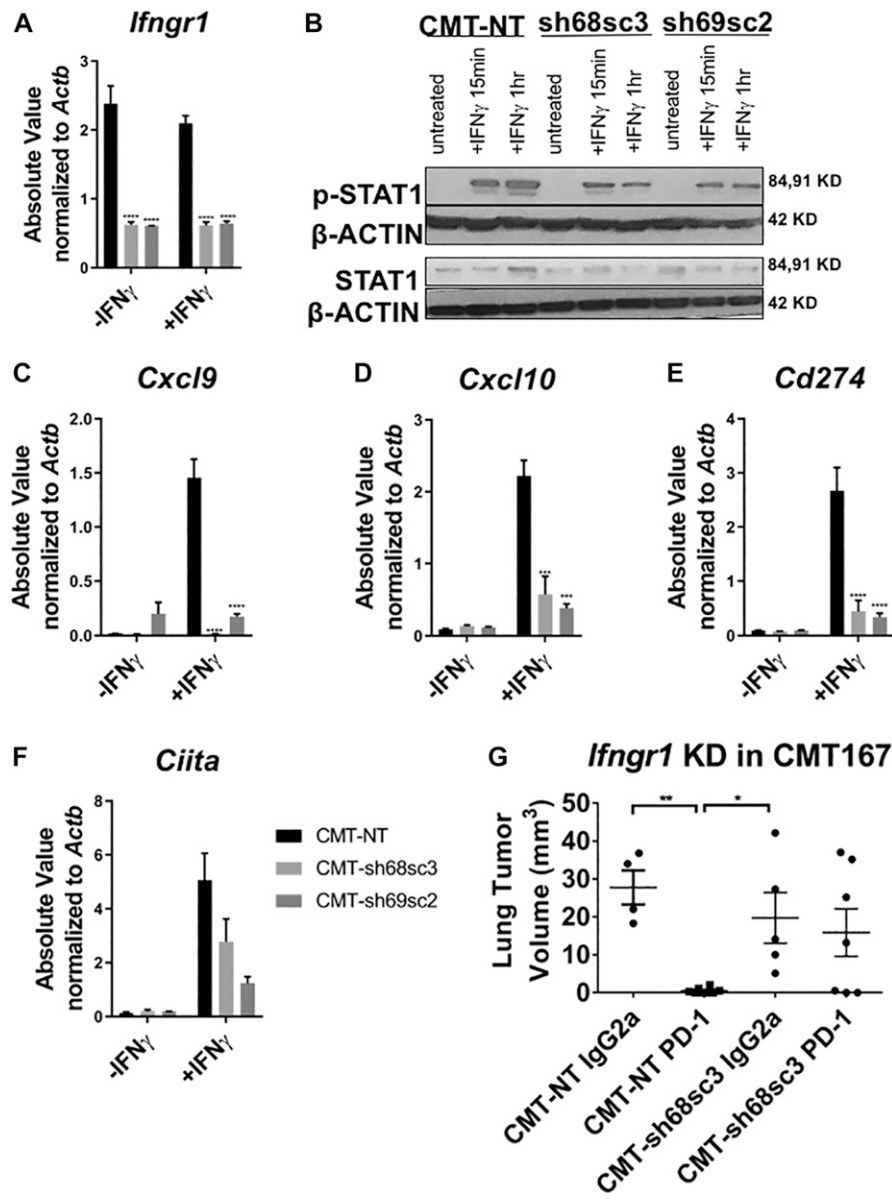

**Figure 3. Silencing *Ifngr1* in the CMT167 line confers decreased response to IFNγ in vitro and in vivo.**
Two separate shRNAs targeting *Ifngr1* (CMT-sh68 and CMT-sh69) and a nontargeting control vector (CMT-NT) were transduced into CMT167 cells expressing luciferase. CMT167 cells were then screened for functional and stable knockdown of *Ifngr1* after 10 d of puromycin selection and subcloning of shRNA pools (CMT-sh68sc3 and CMT-sh69sc2). The cells were treated with ±10 ng/ml IFNγ for 24 h followed by isolation of RNA and qRT-PCR. **(A)** mRNA levels of (A) *Ifngr1* was assessed for knockdown of shRNA subclones relative to the CMT-NT cell line. **(B)** Immunoblots showing p-STAT1, total STAT1, and *β*-ACTIN levels of the CMT-NT, CMT-sh68sc3, and CMT-sh69sc2 cell lines ± IFNγ after 15 min or 1 h in vitro. **(C–F)** mRNA expression of downstream IFNγ response genes (C) *Cxcl9*, (D) *Cxcl10*, (E) *Cd274*, and (F) *Ciita* are shown as absolute values (SQ values) normalized to the housekeeping gene *Actb* after ±10 ng/ml IFNγ treatment for 24 h. Statistics compare the CMT-NT cell line to other cell lines with IFNγ treatment. CMT-NT or CMT-sh68sc3 cells were orthotopically injected into the lungs of syngeneic mice, established for 7 d, then were treated with either an isotype control antibody (IgG2a) or an anti–PD-1 antibody for 3 wk followed by terminal euthanization at 4 wk post tumor cell injection. **(G)** Primary tumor volume was assessed using digital calipers. Error bars represent the mean of the data ± SEM after a two-way ANOVA (A, C–F) or a one-way ANOVA (G) (*P < 0.05, **P < 0.01, ***P < 0.001, and ****P < 0.0001).
Source data are available for this figure.

proliferating cancer cells in LLC-sh21 versus LLC-NT replicates (Figs 5D and S5D, and E). Both LLC-NT and LLC-sh21 cancer cells expressed MHC class I but had low levels of MHC class II expression, suggesting selective interactions with CD8⁺, but not CD4⁺ T cells (Fig S5E). Interestingly, PD-L1 expression was not high on cancer cell clusters in either the LLC-NT or LLC-sh21 groups, indicating that although LLC-sh21 cells induce PD-L1 expression in vitro after IFNγ treatment (Figs S3F and S4C), this is not reflected in vivo (Fig S5E).

In addition to populations of cancer cells, we examined differences in immune cell populations between LLC-NT and LLC-sh21 tumors by examining CD45⁺ cells. We have previously profiled macrophage populations in LLC tumors using conventional flow cytometry (Poczobutt et al, 2016a). We found that a population of recruited macrophages, defined as CD11b⁺/CD11c⁺/CD64⁺/SiglecF⁻ macrophages (cluster #7), was enriched in tumor-bearing lungs

(LLC-NT and LLC-sh21) relative to naive lungs (Fig 5C). In addition, a population of recruited monocytes, defined as CD11b⁺/CD11c⁻/CD64⁺/SiglecF⁻ monocytes (cluster #2), was selectively enriched in LLC-NT tumors relative to LLC-sh21 tumors or naive lungs (Fig 5D). Conversely, a subset of alveolar macrophages defined as CD11b⁻/CD11c⁺/CD64⁺/SiglecF⁺ macrophages (cluster #3), was enriched in LLC-sh21 tumors compared with LLC-NT or naive lungs (Fig 5E). Although not enriched to a significant degree, we also detected increased CD8⁺ and CD4⁺ T cells in LLC-sh21 tumors versus LLC-NT tumors (Fig 5A and B).

Because we observed low levels of PD-L1 on both LLC-NT and LLC-sh21 cancer cells in vivo, we wanted to determine if expression of PD-L1 on non-cancer cells was altered in mice implanted with LLC-sh21 *Socs1*-KD cancer cells (Li et al, 2017). Although the overall frequency of CD64⁺ myeloid cells was comparable between LLC-NT

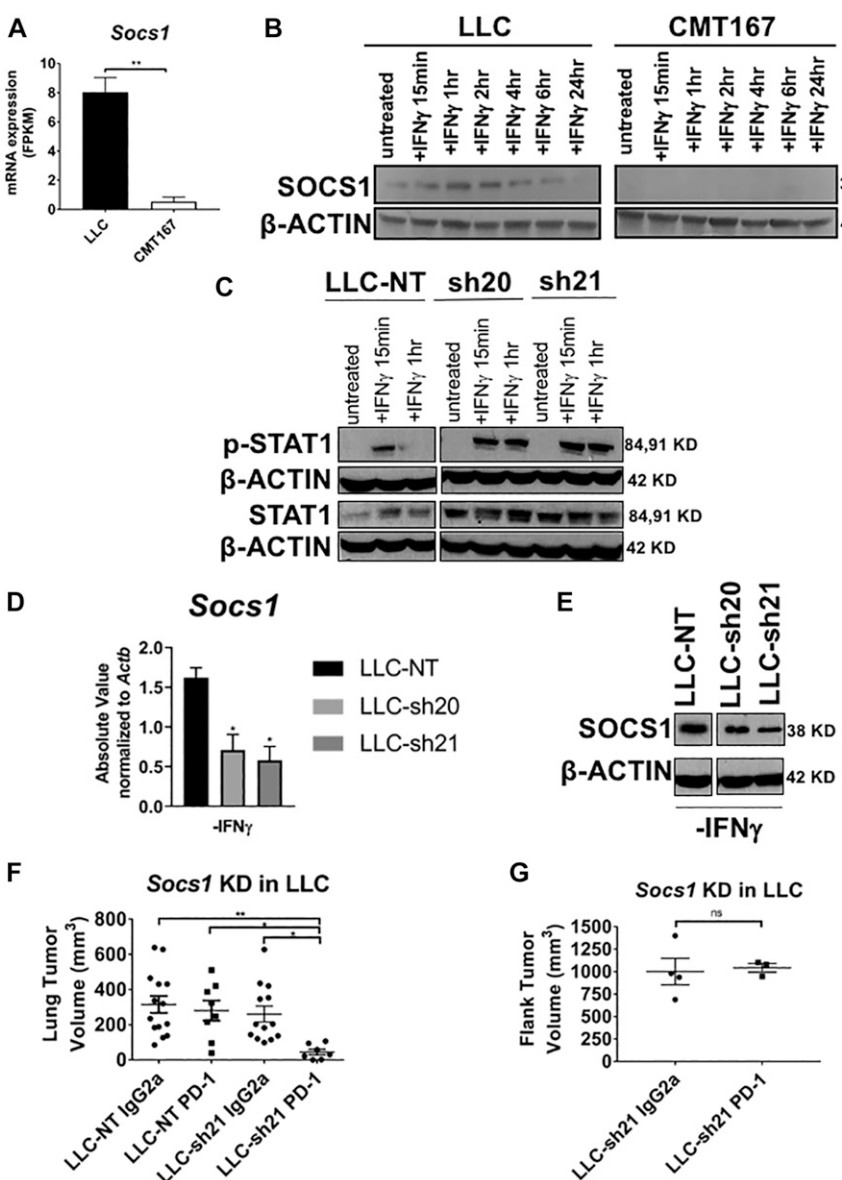

**Figure 4. Silencing *Socs1* in the LLC line confers increased response to IFNγ in vitro and in vivo.**
**(A)** In vitro mRNA expression of *Socs1* in FPKM as assessed by RNA-Seq in LLC and CMT167 cells. **(B)** Immunoblots of LLC or CMT167 cells treated with ±10 ng/ml IFNγ in vitro for a time course ranging from 15 min to 24 h, showing SOCS1 expression relative to the housekeeping gene *β*-ACTIN. Two separate shRNAs targeting *Socs1* (LLC-sh20 and LLC-sh21) and a nontargeting control vector (LLC-NT) were transduced into LLC cells expressing luciferase. LLC cells were then screened for functional and stable knockdown of *Socs1* after 10 d of puromycin selection. **(C)** Immunoblots showing p-STAT1, total STAT1, and *β*-ACTIN levels of the LLC-NT, LLC-sh20, and LLC-sh21 cell lines ± IFNγ at 15 min or 1 h in vitro. **(D)** mRNA expression of *Socs1* via qRT-PCR shown as absolute values (SQ values) normalized to the housekeeping gene *Actb*. Statistics compare the LLC-NT line to the knockdown cell lines. **(E)** Immunoblot for SOCS1 relative to *β*-ACTIN protein levels in the LLC-NT, LLC-sh20, and LLC-sh21 cell lines in vitro at baseline. LLC-NT or LLC-sh21 cells were orthotopically injected into the lungs of syngeneic mice, established for 7 d, then were treated with either an isotype control antibody (IgG2a) or an anti–PD-1 antibody for 2 wk followed by terminal euthanization at 3 wk post tumor cell injection. **(F)** Primary tumor volume of lung tumors was assessed using digital calipers. LLC-sh21 cells were injected into the flanks of syngeneic mice, established for 7 d, then were treated as in (F). **(G)** Primary tumor volume of subcutaneous tumors (Flank) was also assessed with digital calipers. Error bars represent the mean of the data ± SEM after a *t* test (A, G) or a one-way ANOVA (D, F) (*P < 0.05, **P < 0.01, ***P < 0.001, and ****P < 0.0001).
Source data are available for this figure.

and LLC-sh21 tumors, the relative distribution of macrophages expressing variable PD-L1 levels was altered (Fig 6A and B). Recruited monocytes (cluster #2: CD11b⁺/CD11c⁻/CD64⁺/SiglecF⁻ cells) had low levels of PD-L1 expression and were less abundant in LLC-sh21 tumors compared with LLC-NT tumors (Fig 6A–C) (Poczobutt et al, 2016a). Recruited macrophages (cluster #7: CD11b⁺/CD11c⁺/CD64⁺/SiglecF⁻ cells), which express intermediate levels of PD-L1, were unchanged (Fig 6A, B, and D). Resident alveolar macrophages (clusters #3 and #4: CD11b⁻/CD11c⁺/CD64⁺/SiglecF⁺ cells), which have the highest level of expression of PD-L1 relative to the other macrophage subsets, were increased in LLC-sh21 tumors (Fig 6A, B, and E). These data collectively indicate that there is an increase in PD-L1ʰⁱ and a reciprocal decrease in PD-L1ˡᵒ macrophages in LLC-sh21 tumors, which may explain the efficacy of anti–PD-1 on these tumors.

To determine the importance of PD-L1 expression on cells of the TME versus tumor-intrinsic PD-L1, we compared the response of

LLC-sh21 tumors implanted into WT or PD-L1⁻/⁻ mice to anti–PD-1 treatment. When implanted into PD-L1⁻/⁻ mice, LLC-sh21 tumors lose their sensitivity to anti–PD-1 (Fig 6F). These results indicate that the PD-L1 expression in the TME is a major determinant of response to anti–PD-1 treatment.

### *Socs1* KD tumors exhibit a more T cell–inflamed phenotype and alterations in macrophage composition

Although not statistically significant by CyTOF, we observed an increase in CD4 and CD8 T cell populations in LLC-sh21 tumors relative to LLC-NT tumors. Because T cells are critical for responses to immune checkpoint inhibitors, we further characterized changes in T cell populations by immunostaining. Representative images from both an LLC-NT and LLC-sh21 tumor harvested at 2 wk without treatment are shown (Figs 7A and B, S7A, and B). Quantification of

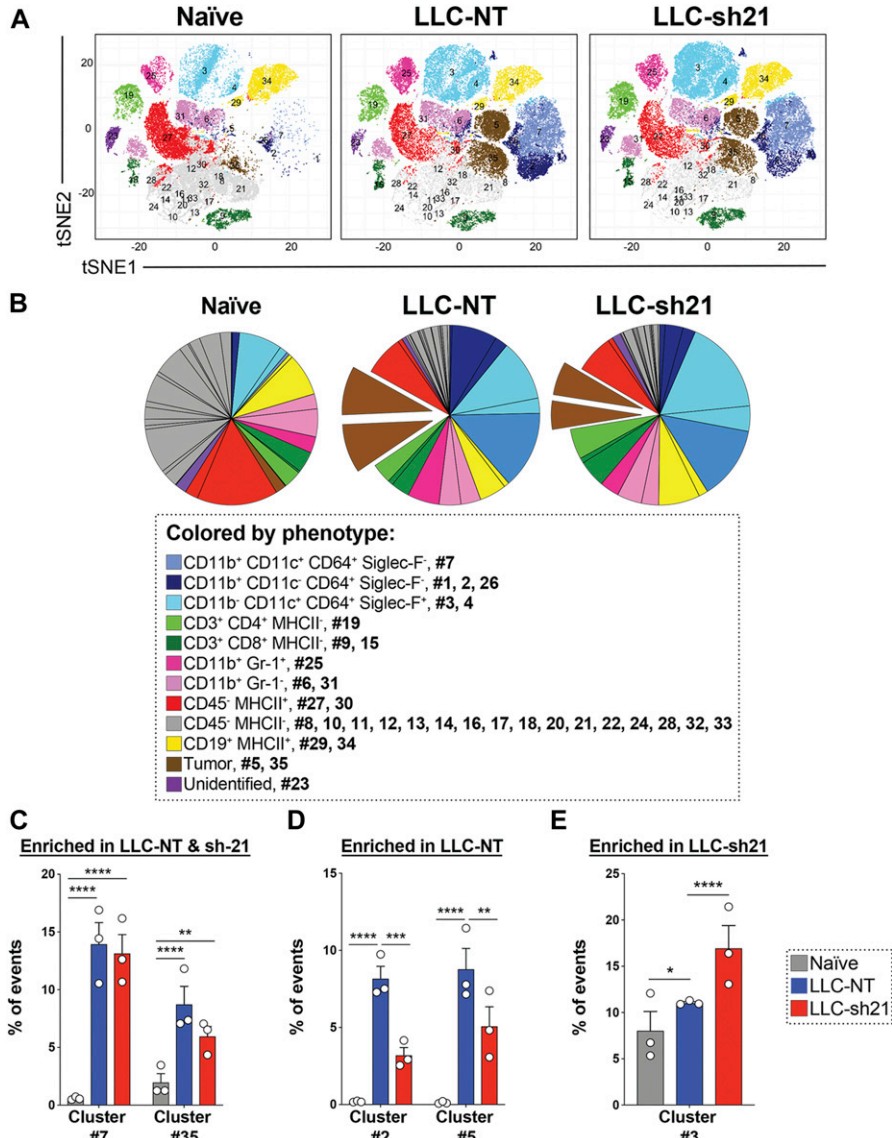

**Figure 5. *Socs1* KD tumors show alterations in multiple populations.**
LLC-NT or LLC-sh21 cells were orthotopically injected into the left lung lobes of mice and established primary tumors. After 2 wk of tumor growth with no treatment, the mice were euthanized and their tumor-bearing lung lobes were isolated. Single-cell suspensions were made from tumor-bearing lung lobes or naïve lungs. Naïve biological replicates each contained lungs isolated from one mouse (three mice/three replicates total). LLC-NT and LLC-sh21 biological replicates each contained tumor-bearing lungs from three pooled mice (nine mice/three replicates total; three mice/replicate). Single-cell suspensions were then stained with a 39-antibody panel and analyzed on the Helios mass cytometer. Data show all viable single cells, subjected to the PhenoGraph algorithm. **(A)** PhenoGraph-defined cellular distribution and clustering, as defined by tSNE1 and tSNE2, colored by phenotypic designation (legend provided in panel B) for all treatment conditions where all replicates per experimental condition are combined. **(B)** Pie charts show all 35 clusters colored by their phenotypic designations for all experimental conditions with numbers indicating which PhenoGraph-defined clusters were present in each phenotypic designation. **(C–E)** Clusters identified as statistically significant are shown as preferentially enriched in (C) both the LLC-NT and LLC-sh21 tumor samples, (D) LLC-NT samples alone, or (E) LLC-sh21 samples alone. Error bars represent the mean of the data ± SEM after a two-way ANOVA (C–E) (*$P < 0.05$, **$P < 0.01$, ***$P < 0.001$, and ****$P < 0.0001$).

cells per high-power field (40× magnification) showed that LLC-sh21 tumors had significant increases in CD3+ T cells, a pan T cell marker, as well as trending increases in CD8+ and CD4+ populations relative to LLC-NT tumors (Figs 7C and D, and S7C), consistent with our CyTOF data.

Changes in T cells were also analyzed by flow cytometry (Fig S7D). By flow, we observed a significant increase in CD8+ T cells expressing PD-1 (Fig 7E) and a trending increase in CD8+/PD-1+/CD69+ T cells in LLC-sh21 versus LLC-NT tumors (Fig 7F), indicating that a significant percentage of PD-1 expressing CD8+ T cells had recently seen antigen. These changes were not observed in CD4+ T cells (Fig S7E and F) Upon cellular stimulation with PMA/Ionomycin in the presence of Golgi inhibitors, we detected significant increases in IFNγ-positive and IFNγ/TNFα double-positive CD8+ T cells in LLC-sh21 tumors versus LLC-NT tumors (Fig 7G and H) indicating more cytotoxic and antitumor capacities. Similar increases were observed in CD4+ T cells, although these were not statistically significant (Fig S7G and H). These data indicate that LLC-sh21 tumors at baseline, before anti–PD-1 therapy, have greater CD8+ T cell activation and by extension, recognition of tumor cells than LLC-NT tumors.

### *Socs1* KD tumors have elevated levels of Cxcl9

Because we observed increased tumor-infiltrating T cells in LLC-sh21 *Socs1* KD tumors, as well as increased CD8+ T cell activation, we sought to identify cancer cell–intrinsic factors that could mediate these effects. We, therefore, recovered LLC-NT and LLC-sh21 cancer cells implanted into GFP-expressing transgenic mice and compared gene expression profiles with the respective cancer cells grown in vitro using RNA-Seq (as in Fig 1A with parental LLC and CMT167 cells). These data showed that *Socs1* expression was decreased by ~60% in LLC-sh21 compared with LLC-NT cancer cells in vivo, confirming that these cells were still silenced for *Socs1* (Table S4).

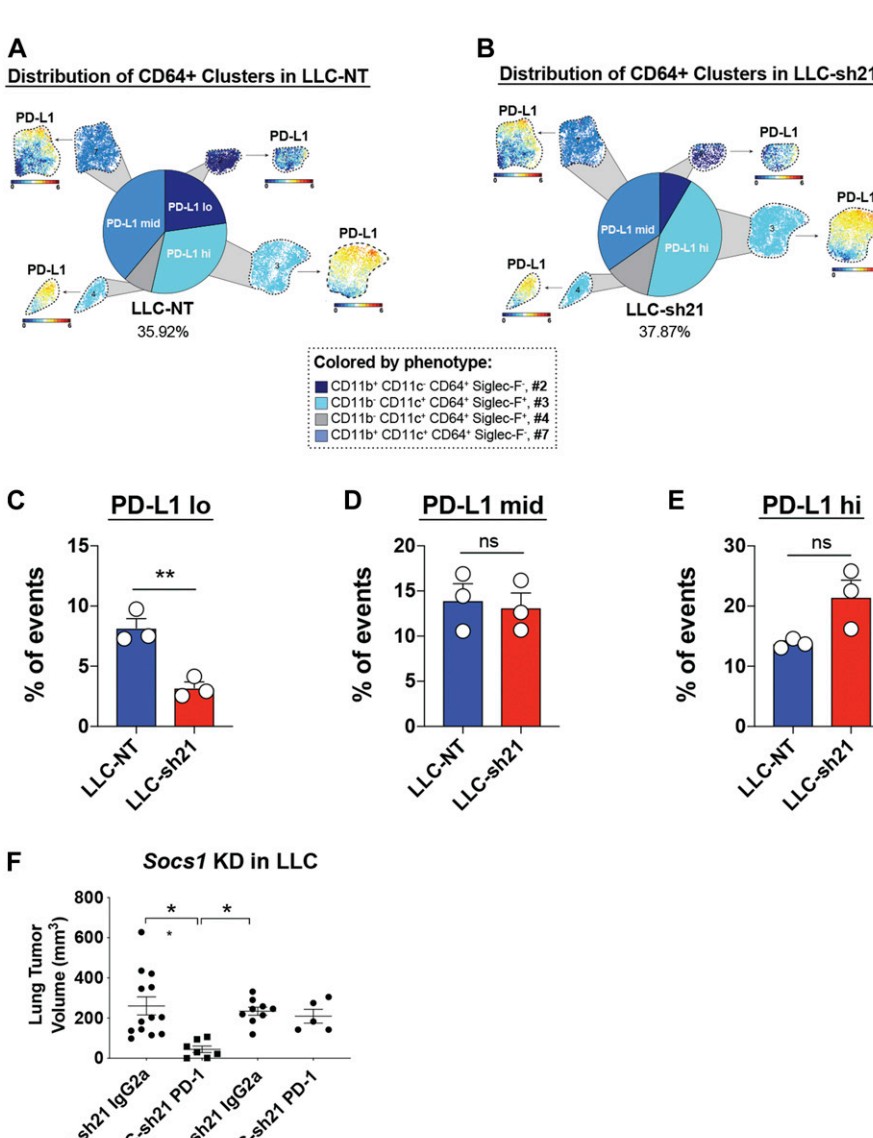

**Figure 6. Socs1 KD tumors have an altered macrophage composition.**
LLC-NT or LLC-sh21 cells were orthotopically injected into the left lung lobes of mice and established primary tumors. After 2 wk of tumor growth with no treatment, mice were euthanized and their tumor-bearing lung lobes were isolated. Single-cell suspensions were made from tumor-bearing lung lobes or naïve lungs. Naïve biological replicates each contained lungs isolated from one mouse (three mice/three replicates total). LLC-NT and LLC-sh21 biological replicates each contained tumor-bearing lungs from three pooled mice (nine mice/three replicates total; three mice/replicate). Single-cell suspensions were then stained with a 39-antibody panel and analyzed on the Helios mass cytometer. Data show all viable single cells, subjected to the PhenoGraph algorithm. **(A, B)** Pie charts show the relative frequency of cell clusters containing CD64⁺ events (a pan macrophage marker) in (A) LLC-NT or (B) LLC-sh21 tumors. Clusters are colored according to PD-L1 expression. **(C–E)** Pie charts are quantified based on (C) PD-L1 lo (cluster #2), (D) PD-L1 mid (cluster #7), or (E) PD-L1 high (clusters #3 and #4) expression. LLC-sh21 cells were orthotopically injected into the lungs of either WT or PD-L1$^{-/-}$ mice, established for 7 d, then were treated with either an isotype control antibody (IgG2a) or an anti–PD-1 antibody for 2 wk followed by terminal euthanization at 3 wk post tumor cell injection. **(F)** Primary tumor volume was assessed via digital calipers. Error bars represent the mean of the data ± SEM after a t test (C–E) or one-way ANOVA (F) (*$P < 0.05$, **$P < 0.01$, ***$P < 0.001$, and ****$P < 0.0001$).

We identified 44 genes that were differentially expressed between the LLC-NT and LLC-sh21 cells in vivo (Table S4) that met our strict q-value criteria of q < 0.05. Of interest, expression of *Cxcl9* as well as three MHC Class I genes (*H2k1*, *H2q1*, and *H2q4*) was increased in LLC-sh21 cells in vivo compared with the LLC-NT cells (Fig 8A and Table S4). We confirmed changes in *Cxcl9* by in situ hybridization looking at the tumor edge, where many infiltrating immune cells can be found, versus the middle of tumors, where it is more difficult for these cells to infiltrate (Fig 8B). Relative to tumor sections stained with a negative control probe, dapB (Fig 8C), or a normal adjacent lung stained with a probe for murine *Cxcl9* (another negative control), (Fig 8C) four separate LLC-NT or LLC-sh21 tumors stained positive for *Cxcl9* (Fig 8D and E). Interestingly, there was much higher *Cxcl9* staining in three of four LLC-sh21 tumors, particularly around the tumor edge compared with LLC-NT tumors.

Although we do not exclude other potential tumor-intrinsic mechanisms, the increased levels of *Cxcl9* within and around LLC-sh21 tumors would allow for increased T cell infiltration and trafficking into tumors.

## Discussion

Although biomarkers have been developed, which correlate the response of lung cancer patients to anti-PD1/anti–PD-L1 therapy, defining the cellular and molecular pathways that regulate this response remains poorly understood. We have previously demonstrated differential responses to anti–PD-1/anti–PD-L1 of two K-Ras–mutant lung cancer cells, with CMT167 tumors showing a strong inhibition and LLC tumors being resistant to therapy (Li et al,

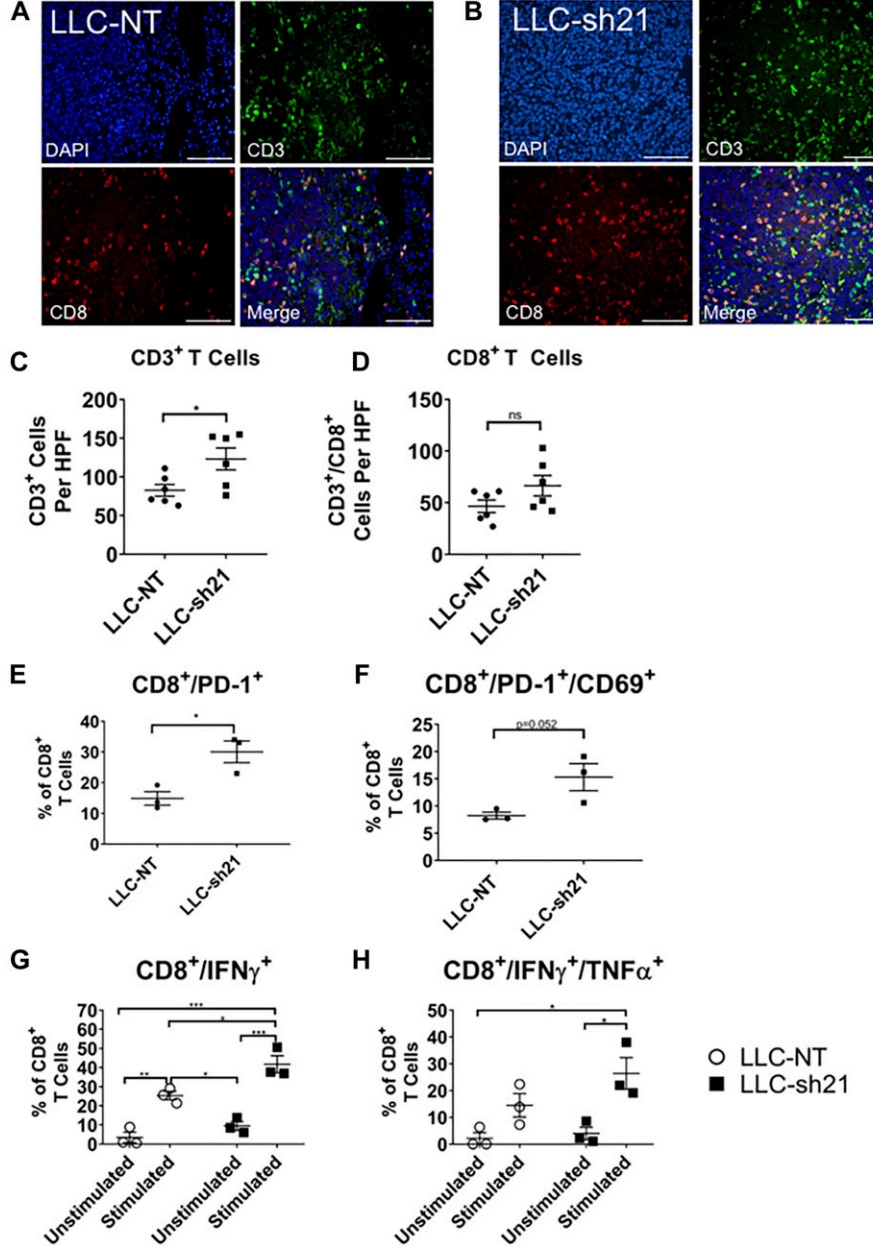

**Figure 7. Socs1 KD tumors exhibit a more T cell–inflamed phenotype and increased CD8⁺ T cell activation.**
LLC-NT or LLC-sh21 cells were orthotopically injected into the left lung lobes of mice and established primary tumors. After 2 wk of tumor growth with no treatment, the mice were euthanized and their tumor-bearing lung lobes were isolated for either flow cytometry or FFPE and T cell staining by immunofluorescence. **(A, B)** Representative images of T cell staining from (A) an LLC-NT tumor or (B) an LLC-sh21 tumor (scale bars: 100 μm), showing CD8⁺ T cell staining. DAPI is shown in blue, CD3 in green, CD8 in red, and a merge of all channels in yellow. **(C, D)** Quantification of CD3⁺ T cells and (D) CD8⁺ T cells per high-power field (HPF) in LLC-NT versus LLC-sh21 tumors. There were six tumors from LLC-NT mice, and six tumors from LLC-sh21 mice. T cell numbers per HPF were averaged over four experiments (six random fields per tumor × four staining experiments = 24 fields averaged/tumor in total). Quantification of T cells was performed by two blinded observers (BB & AN). For flow cytometry, single-cell suspensions were made from tumor-bearing lung lobes. There were three experimental replicates × three tumor-bearing pooled lung lobes, for a total of nine lung lobes per the experimental conditions of "LLC-NT" or "LLC-sh21". **(E, F)** For the "T Cell Phenotypic Panel," single-cell suspensions were assessed for the following: (E) the percentage of PD-1–expressing CD8⁺ T cells or (F) double-positive PD-1/CD69–expressing CD8⁺ T cells gated as a percentage of all CD8⁺ T cells. For the "T Cell Stimulation Panel," single-cell suspensions were stimulated with brefeldin A, monensin, and a cellular stimulation cocktail (PMA/Ionomycin) for 5 h to determine intracellular cytokine production at the time of harvest. Cell suspensions were either unstimulated (treated with brefeldin and monensin alone) or stimulated (treated with brefeldin, monensin, and PMA/Ionomycin). **(G, H)** Single-cell suspensions were assessed for the following: (G) the percentage of single-positive IFNγ-expressing CD8⁺ T cells or (H) double-positive IFNγ/TNFα–expressing CD8⁺ T cells gated as a percentage of all CD8⁺ T cells. Error bars represent the mean of the data ± SEM after a t test (C–F) or a two-way ANOVA (G–H) (*P < 0.05, **P < 0.01, ***P < 0.001, and ****P < 0.0001).

2017). In this study, we have sought to define intrinsic features of the cancer cell that mediate this differential response. Our data define responsiveness of cancer cells to IFNγ as a critical determinant for sensitivity to anti–PD-1 therapy. Furthermore, we have shown that altering responsiveness of the cancer cells to IFNγ causes complex multifaceted changes in the microenvironment.

By analyzing gene expression changes in vivo, we determined that LLC cells failed to robustly induce an IFNγ signature compared with CMT167 tumors. An IFNγ signature in bulk tumor tissue (which includes a complex mixture of cancer cells, stromal cells and immune cells etc.) has been associated with responsiveness to anti–PD-1 therapy in lung cancer and others malignancies (Ayers et al, 2017). However, whether this is associated with cancer cells

alone or the surrounding TME has not been well examined. We hypothesized that sensitivity of cancer cells to IFNγ is a major regulator of the TME, and that altering the sensitivity of the cancer cells would regulate the response of tumors to checkpoint inhibition.

Regarding cancer-cell–intrinsic SOCS1 expression, previous studies have identified *Socs1* as a gene associated with immuno-suppression and tumor progression. In human tumors, MET activation was associated with increased SOCS1 expression and escape from immunotherapy (Saigi et al, 2018). At the same time, chemotherapeutic agents were shown to down-regulate *Socs1* through induction of *miR-155*, resulting in increased activation of CD8⁺ T cells (Ye et al, 2018). In a mouse model of melanoma, an in vivo

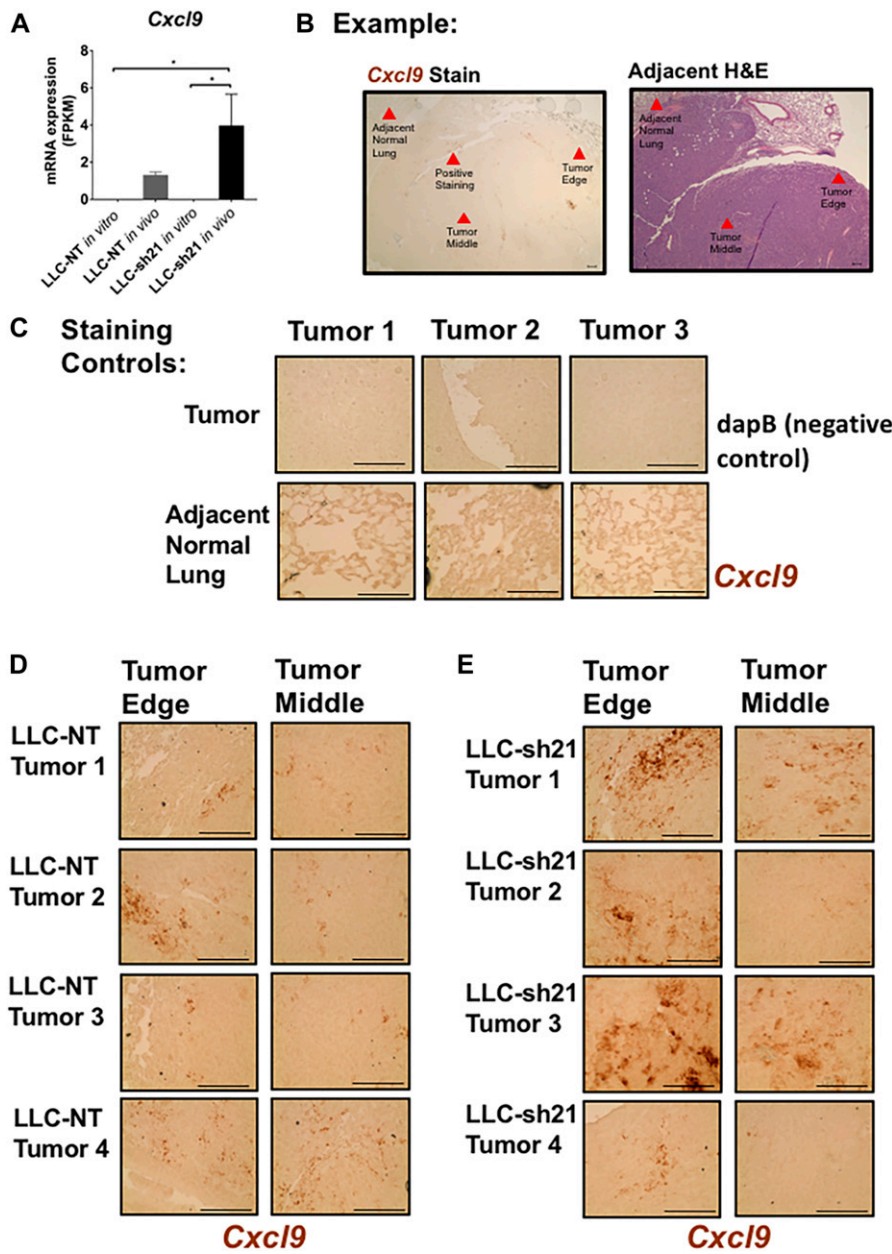

**Figure 8. *Socs1* KD tumors have elevated levels of *Cxcl9*.**
LLC-NT or LLC-sh21 cells were orthotopically injected into the left lung lobe of transgenic GFP-expressing C57BL/6J mice and were grown for 3 wk with no treatment. Tumor-bearing lung lobes were isolated and made into single-cell suspensions containing both GFP-positive (host cells) and GFP-negative (cancer cells). First, RNA was isolated from identical cancer cells grown in passage (in vitro condition). Second, RNA was isolated from recovered GFP-negative cancer cells (isolated via FACS-in vivo condition). Third, RNA was run for RNA-Seq from both conditions. Both the LLC-NT and LLC-sh21 conditions had three experimental replicates per in vitro and in vivo conditions with five tumor-bearing lung lobes pooled per in vivo experimental replicate (15 mice used total). **(A)** In vitro and in vivo mRNA expression of (A) *Cxcl9* in FPKM as assessed by RNA-Seq. The RNAScope system was used for in situ hybridization. **(B)** Example of positive tumor staining with a murine *Cxcl9 probe* and its adjacent H&E stain (scale bars: 10 $\mu$m). Red arrows denote the tumor edge, tumor middle, adjacent normal lung, or positive *Cxcl9* staining. **(C)** Staining controls of (C) tumor tissue stained with a negative control probe, dapB, or adjacent normal lung stained with a murine *Cxcl9* probe that both show negative staining (scale bars: 100 $\mu$m). **(D, E)** Four separate tumors from either the LLC-NT or (E) LLC-sh21 experimental conditions stained with a probe targeting murine *Cxcl9* (scale bars: 100 $\mu$m). Dark brown regions represent positive Cxcl9 staining. Error bars represent the mean of the data ± SEM after a one-way ANOVA (A). (*$P < 0.05$, **$P < 0.01$, ***$P < 0.001$, and ****$P < 0.0001$). Source data are available for this figure.

CRISPR screen identified novel immunotherapy targets, including PTPN2 and SOCS1, both of which act to decrease IFN signaling. Knockout of *Ptpn2* in formerly resistant cancer cells resulted in responsiveness to immunotherapy, similar to our LLC-sh21 cells silenced for *Socs1*. Thus, we speculate that therapies targeting the SOCS family of proteins and similar phosphatases to PTPN2 could be used in combination with immunotherapy.

Tumors can evade the immune system by either adaptive or acquired resistance, which both involve IFNγ signaling (Pardoll, 2012). First, if cancer cells have the capacity to respond to IFNγ, they will often up-regulate immune checkpoints on their surface, such as PD-L1, and will effectively shut down effector T cell responses (Pardoll, 2012). This phenomenon is known as adaptive immune resistance and represents a therapeutic vulnerability for tumors

similar to CMT167 tumors (Pardoll, 2012; Juneja et al, 2017). In these cells, silencing the expression of *Ifngr1* decreased JAK/STAT signaling and made tumors resistant to anti–PD-1 therapy with similar results obtained in melanoma models (Gao et al, 2016). This effect does not appear to be a consequence of altered proliferation of the silenced cells but was associated with decreased infiltration of T cells into the tumors, consistent with the model proposed that less-inflamed tumors are resistant to immunotherapy (Gajewski et al, 2006; Spranger et al, 2016). However, because only one shRNA targeting *Ifngr1* was used in vivo, we cannot rule out off-target effects in these studies.

Second, cancer cells can acquire mutations in the IFNγ signaling pathway that hinder their ability to respond to IFNγ, otherwise known as acquired resistance (Pardoll, 2012; Benci et al, 2016; Gao

et al, 2016; Manguso et al, 2017; Shin et al, 2017; Sucker et al, 2017). This phenomenon is similar to what we observe in LLC tumors, which are not therapeutically vulnerable to checkpoint inhibition. Although LLC cells do not harbor JAK/STAT or SOCS1 mutations, they have a diminished response to IFNγ in vitro. This was associated with high basal expression of SOCS1, and silencing *Socs1* markedly increases LLC cells' response to IFNγ in vitro through enhanced JAK/STAT signaling. Importantly, cancer cells with *Socs1* silencing (LLC-sh21 cells) grown as orthotopic tumors are much more responsive to anti–PD-1 therapy than LLC-NT control tumors. This is not observed if LLC-sh21 cells are implanted subcutaneously, suggesting that silencing *Socs1* alters communication between cancer cells and specific components of the lung microenvironment that are not present in subcutaneous tumors. As above, we cannot preclude off-target shRNA effects in these studies.

Our data indicate that LLC-sh21 tumors exhibit tumor extrinsic changes compared with LLC-NT tumors of equal size. We observed a decrease in the percentage of cancer cells in LLC-sh21 compared with LLC-NT tumors and a reciprocal increase in T cells, which confirmed our immunostaining data. We also observed complex changes in in the myeloid compartment, with proportional increases in PD-L1hi resident alveolar macrophages and decreases in PD-L1lo and mid-expressing recruited monocytes/macrophages. In essence, LLC-sh21 tumors exhibit the direct opposite myeloid phenotypes than those observed in parental LLC and LLC-NT tumors. Thus, in LLC-sh21 tumors where cancer cells are silenced for *Socs1*, there is an increased proportion of PD-L1hi macrophages, with the greatest contribution of PD-L1 expression coming from resident alveolar macrophages.

We have previously characterized myeloid populations in the lungs of mice implanted with LLC cells compared with naive mice (Poczobutt et al, 2016a). Multiple populations were recovered using FACS and analyzed using RNA-Seq. These included resident alveolar macrophages (CD11c$^+$/CD11b$^-$/SigF$^+$/Ly6C$^-$/MHCII$^{lo}$), a mix of monocytes/macrophages and CD11b$^+$ dendritic cells (CD11b$^+$/CD64$^{lo}$ CD11c$^+$/Ly6C$^{lo}$/MHCII$^{-/+}$), recruited monocytes (CD11b$^+$/CD64$^{mid}$/CD11c$^-$/Ly6C$^{hi}$/MHCII$^{-/lo}$), and finally recruited macrophages (CD11b$^+$/CD64$^{hi}$/CD11c$^+$/Ly6C$^{-/lo}$/MHCII$^{mid/hi}$). During tumor progression, there was a decrease in *Cd274* (PD-L1)$^{hi}$ alveolar macrophages as assessed by RNA-Seq and a reciprocal increase in recruited monocytes and macrophages which expressed low to intermediate levels of *Cd274* (Poczobutt et al, 2016a). Our CyTOF data with LLC-NT control tumors recapitulated the RNA-Seq data of parental LLCs. Importantly, the same subsets of myeloid cells were found using an unbiased clustering method to analyze our CyTOF data, as those that were identified via RNA-Seq.

Although we have not directly assessed why LLC-sh21 tumors have decreased cancer cell burden relative to LLC-NT control tumors before immunotherapy, we believe that there are several compensatory mechanisms occurring that could explain these differences. One of these mechanisms can likely be attributed to increased *Cxcl9* expression in LLC-sh21 tumors, which we detected by message in tumor sections. Because of elevated *Cxcl9* expression in these tumors, there were more recruited and activated T cells—particularly CD8$^+$ T cells within and around tumors. These results are indicative of a more "T cell–inflamed tumor," which is associated with better response to checkpoint inhibitors (Spranger et al, 2016).

We also anticipated increased expression of PD-L1 on LLC-sh21 cancer cells in vivo, yet cancer cells recovered from tumors only showed a modest induction of PD-L1 compared with LLC-NT cells (Table S4). Thus, the therapeutic vulnerability of LLC-sh21 to anti–PD-1 treatment is unlikely due to the adaptive resistance of cancer cells. However, these data could be explained by only a subset of the cancer cells expressing PD-L1, for instance, at the tumor edge. In our previous studies, PD-L1 expression on CMT167 cells was not detected on all cancer cells, despite their responsiveness to IFNγ and anti–PD-1 therapy (Li et al, 2017). This study also demonstrated that a critical difference between CMT167 and LLC tumors was markedly increased numbers of PD-L1hi macrophages on CMT167 tumors (Li et al, 2017). Therefore, silencing *Socs1* has converted LLC tumors to have many of the features of CMT167 tumors: increased T cell recruitment likely because of increased levels of CXCL9 and increased numbers of PD-L1hi macrophages.

Consistent with this model, PD-L1 host expression was critical for response to immunotherapy in LLC-sh21 tumors (Fig 6F). Another factor that may account for the alterations of myeloid cell populations is decreased expression of *Ccl2* by LLC-sh21 cancer cells recovered from tumors (Table S4). CCL2 is a cytokine that recruits macrophages and monocytes to sites of inflammation. Decreased *Ccl2* expression in LLC-sh21 tumors may account for the decreased frequencies of these recruited macrophage/monocyte populations—which highly express CCL2's cognate receptor, CCR2 by RNA-Seq (Poczobutt et al, 2016a). Future studies will be required to determine if knockout or knockdown of *Ccl2* in parental LLC cells would skew the TME to one with less recruited monocytes and macrophages and a higher proportion of PD-L1hi alveolar macrophages as is seen in LLC-sh21 *Socs1* KD tumors. It would also be of interest to use pharmacological approaches such as commercially available CCL2/CCR2 targeted inhibitors to treat parental LLC tumors. In this way, we would be able to assess the unique contributions of myeloid subsets on tumor progression and response to single-agent immunotherapy.

Unfortunately, chronic stimulation of cancer cells with IFNγ can lead to adaptive or acquired resistance (Benci et al, 2016). Although our results might appear to contradict these findings, there is a possibility that if we continued to treat LLC-sh21 tumors with anti–PD-1 therapy, they might eventually develop resistance through selective pressure on cancer cells to adapt against immune attack, chronic T cell exhaustion, or skewing of immune cells to more immunosuppressive phenotypes (Alspach et al, 2019). However, our results suggest that the early TME of LLC-sh21 tumors is the most therapeutically vulnerable because of an increased percentage of PD-L1hi macrophages, specifically resident alveolar macrophages at this time. Finally, we propose that the lack of alveolar macrophages (PD-L1hi) in the subcutaneous model are responsible for the lack of response of subcutaneous LLC-sh21 tumors to anti–PD-1 therapy.

In summary, our data identify a critical role for IFNγ sensitivity within cancer cells as a major determinant that directly shapes the TME. The results also underscore the complex interplay between cancer cells and populations of inflammatory and immune cells. These interactions are mediated through production of paracrine factors, including chemokines and potentially lipid mediators. Subtle changes, such as altering expression of a single gene in the cancer cells, change these interactions in profound ways, likely by

altering the secretome of the cancer cells. Therapeutically, in this case, we have generated a tumor with increased T cells and fewer myeloid cells, which is associated with an increased response to anti–PD-1. However, the complexity of the crosstalk suggests that a better understanding of how the various cell populations interact is needed to design more effective combination therapies for treatment of lung cancer.

# Materials and Methods

### Cells

Murine LLC cells expressing firefly luciferase were purchased from Caliper Life Sciences and maintained in DMEM (#10-017-CV; Corning) supplemented with 10% FBS, penicillin/streptomycin, and G418 (500 ng/ml). LLC cells harbor a heterozygous K-Ras$^{G12V}$ mutation (Li et al, 2017). CMT167 cells (gift of Dr. Alvin Malkinson, University of Colorado) were transduced with firefly luciferase and maintained in DMEM (#10-017-CV; Corning) with 10% FBS, penicillin/streptomycin, and G418 (500 ng/ml) (Weiser-Evans et al, 2009). CMT167 cells harbor a K-Ras$^{G12C}$ mutation (Li et al, 2017). Cell lines were confirmed mycoplasma negative every 2 wk and were last tested in January 2019 (#LT07-703; Lonza). To maintain cellular phenotypes and to prevent cross-contamination of murine cell lines, the cells were grown in vitro for less than 10 passages, and for only 2–3 wk before use in in vivo experiments. Cell phenotypes were regularly assessed via proliferation assays and EMT status. No phenotypic changes were observed during the course of these studies.

### Mice and tumor models

Wild-type C57BL/6J and GFP-expressing mice [C57BL/6J-132Tg(UBC-GFP)30Scha/J] were obtained from Jackson Laboratory. Dr. Haidong Dong (Mayo Clinic) provided PD-L1 KO mice on a C57BL/6 background. Experiments were performed on 8–16-wk-old male and female mice. All mice were bred and maintained in the Center for Comparative Medicine at the University of Colorado Anschutz Medical Campus in accordance with established IACUC, U.S. Department of Health and Human Services Guide for the Care and Use of Laboratory Animals, and the University of Colorado Anschutz Medical Campus guidelines. For orthotopic lung tumors, an incision was made on the left lateral axillary line at the xyphoid process level, followed by removal of subcutaneous fat (Poczobutt et al, 2013). Tumor cells were suspended in 1.35 mg/ml Matrigel and Hank's buffered saline solution (1 × 10$^5$ cells-LLC tumors; 5 × 10$^5$ cells-CMT167 tumors in 40 µl/injection) and injected into the left lung lobe through the rib cage with a 30-gauge needle (Weiser-Evans et al, 2009). For subcutaneous tumor cell implantation, animals were implanted with 1 × 10$^6$ cells in the flank.

### Lentiviral transduction and stimulation with IFNγ

Murine shRNA constructs were obtained from Sigma-Aldrich via the University of Colorado Functional Genomics Shared Resource (TRC1): Nontargeting control (SHC001V); shRNAs targeting Socs1: LLC-sh20 (TRCN0000067420), LLC-sh21 (TRCN0000067421); shRNAs

targeting Ifngr1: CMT-sh68 (TRCN0000067368), and CMT-sh69 (TRCN0000067369). LLC or CMT167 cells were transduced with lentiviral particles generated from HEK293T cells transfected with shRNA vectors and lentiviral helper plasmids. Viral supernatant was collected at both 24 and 48 h after transfection. Before transduction, LLC and CMT167 cells were pretreated with polybrene for 1 h. During this time, polybrene was also added to viral supernatant generated from HEK293T cells and was filtered through a 0.45 µm filter before media was placed on LLC or CMT167 cells. Stable cells were then selected for after 10 d of puromycin treatment (2 µg/ml). Pools of transduced cells were screened for degree of knockdown by mRNA and protein relative to parental cell lines and to the nontargeting control cells. For CMT167 transduced cells, knockdowns were subcloned and are subsequently represented as "CMT-sh68sc3" or "CMT-sh69sc2." For in vitro experiments, the cells were treated with recombinant murine IFNγ (10–100 ng/ml) (PeproTech #315-05), followed by isolation of protein and/or RNA for immunoblotting and qRT-PCR.

### Immunoblotting

Cells were washed 3× with PBS, followed by lysis with MAPK buffer and a protease inhibitor cocktail from Sigma-Aldrich #P8340, 50 mM β-glycerophosphate, pH 7.2, 0.5% Triton X-100, 5 mM EGTA, 100 µM sodium orthovanadate, 1 mM dithiothreitol, and 2 mM MgCl$_2$. 10–40 µg of total protein was fractionated by SDS–polyacrylamide gel electrophoresis and transferred to Polyvinyledine difuloride membranes. Antibodies used were as follows: pSTAT1 (Y701) Cell Signaling #9167S (1:500–1:1,000); STAT1, Cell Signaling #9172S (1:1,000–1:1,500); SOCS1, Abcam #ab3691 (1:300–1:500); IFNGR1 (interferon γ receptor α), Lifespan Biosciences #LS-C33-4260 (1:300–1:500); IFNGR2 (interferon γ receptor β/AF-1), Abcam #77246 (1:300–1:500); β-ACTIN, Sigma-Aldrich #A5441 (1: 5,000–1:10,000); Rabbit HRP, Jackson Immuno Research #111-035-144 (1: 5,000–1:10,000); and Mouse HRP, Santa Cruz #sc-2005 (1: 5,000–1:10,000).

### Quantitative real-time-PCR

Total RNA from cultured cells was isolated using the RNeasy Mini kit (QIAGEN), followed by reverse transcription with 1 µg of total RNA/sample (qScript cDNA Synthesis kit; QuantaBio). qRT-PCR was conducted on the myIQ Real-Time PCR Detection System (Bio-Rad) using Power SYBR Green PCR Master Mix (Applied Biosystems). Relative message levels of each gene were normalized to the housekeeping gene, Actb or Gapdh (shown as Absolute Values, or Starting Quantity [SQ] Values). For each gene assessed, there were three technical and three experimental replicates. Primers used were as follows: Murine Socs1, F:5′-CTGCGGCTTCTATTGGGGAC-3′, R: 3′-AAAAGGCAGTCGAAGGTCTCG-5′; Murine Cxcl9, F:5′-GAGCAGTGTG-GAGTTCGAGG-3′, R:3′-TCCGGATCTAGGCAGGTTTG-5′; Murine Ciita, F:5′-TGCGTGTGATGGATGTCCAG-3′, R: 3′-CCAAAGGGGATAGTGGGTGTC-5′; Murine Cxcl10, F:5′-GGATGGCTGTCCTAGCTCTG-3′, R:3′-TGAGCTAGGGAG-GACAAGGA-5′; Murine Ifngr1, F: 5′-TACAGGTAAAGGTGTATTCGGGT-3′, R:3′-ACCGTGCATAGTCAGATTCTTTT-5′; Murine Cd274 (PD-L1),F:5′-TGCTGCA-TAATCAGCTACGG-3′, R:3′-GCTGGTCACATTGAGAAGCA-5′; Murine Actb, F: 5′-GGCTGTATTCCCCTCCATCG-3′, R: 3′-CCAGTTGGTAACAATGCCATG-5′; and Murine Gapdh F:5′-CGTGGGAGTCTACTGGTGTCTTC-3′, 5′-CGGAGATGATGACCCTTTTGGC-3′.

## Cxcl9 and Cxcl10 ELISAs

Tumor cells were treated for 48 h ± 100 ng/ml IFNγ in vitro. The medium was collected, spun down to remove floating cells, and ELISA was performed on supernatant according to manufacturer's protocol. 50 µl of conditioned medium was used per replicate. ELISAs: R&D Systems #DY492 and #DY466.

## Cancer cell flow cytometry

Tumor cells were treated in vitro for 18–72 h ± 10–100 ng/ml IFNγ and ±1 µM ruxolitinib (#R-6688; LC Laboratories). The cells were trypsinized or scraped, washed with PBS, and resuspended in an antibody solution. Flow cytometry was performed on the Yeti or Gallios instruments and analyzed using Kaluza software as part of the University of Colorado Cancer Center Flow Cytometry Core. Antibodies and reagents used were as follows: Foxp3 Staining Buffer Set, eBioscience #00-5523-00; Anti-Mouse PD-L1-PE, eBioscience #12-5982-81 (1:200); Ghost 510 Viability Dye, Tonbo Biosciences #13-0870-T100 (1:200); Aqua Viability, Thermo Fisher Scientific #L34957 (1:200); V500 Rat anti-mouse CD4, BD #560782; Anti-Mouse PerCP I-A/I-E (MHCII), BioLegend #107624 (1:200); MHCI (H2-D), eBioscience #17-5998-80 (1:200); MHCI (H2-K), eBioscience #17-5958-80 (1:200); VersaComp Antibody Capture Bead Kit, Beckman Coulter #B22804; and Murine Fc Block, eBioscience #14-0161-86 (1:100).

## RNA isolation from tumor homogenate

Tumor-bearing lung lobes were isolated from mice harboring LLC or CMT167 tumors grown for 2 or 3 wk. Lung lobes were snap-frozen in liquid nitrogen. Upon first thaw, tumor-bearing lung lobes were homogenized using an overhead stirrer (Wheaton) followed by RNA isolation and qRT-PCR as above.

## Anti–PD-1 treatment

Tumor-bearing mice were intraperitoneally injected twice weekly with either an IgG2a isotype control antibody, or an anti–PD-1 antibody (BioXCell) at a dose of 200 µg in PBS per dose (8–10 mg/kg): Anti-Mouse IgG2a, BioXCell #BE0089 Clone 2A3 and Anti-Mouse PD-1, BioXCell #BE0146 Clone RMP1-14.

## Immunofluorescence

Tumor-bearing lungs were perfused with 20 U/ml of PBS/heparin followed by inflation, and then were fixed overnight in 10% formalin and maintained in 70% ethanol until paraffin embedding. 4-µm-thick sections cut from Formalin-Fixed Paraffin-Embedded (FFPE) tissue blocks were deparaffinized, rehydrated, and stained with 0.1% Sudan Black B (Sigma-Aldrich) in 70% ethanol. The slides were heated in a citrate antigen retrieval solution for 2 h at 100°C and quenched with 10 mg/ml sodium borohydride. The slides were blocked with a mixture of goat serum, SuperBlock (SkyTek Laboratories), and 5% BSA overnight. The slides were incubated with primary antibodies in a 1:1 mixture of 5% BSA and SuperBlock for 1 h, followed by incubation with secondary antibodies for 40 min. Slides were coverslipped with Vectashield with DAPI. Hematoxylin and eosin (H&E) stains were performed on one section per tumor by the University of Colorado Denver's Histology Shared Resource Core. For quantitation of T cells, at least three nonserial tumor sections per animal (six animals per experimental condition) were examined. The mean number of $CD3^+/CD4^+/CD8^+$ T cells was obtained from the average of six random 40× tumor fields per section using two blinded observers (BB & AN). Antibodies/reagents used were as follows: Anti-Mouse CD3e, Thermo Fisher Scientific #MA5-14524 Clone SP7 (1:100); Anti-Mouse CD4, eBioscience #14-9766-82 (1:50); Anti-Mouse CD8, eBioscience #14-0808-82 (1:100); AF594 goat anti-rabbit IgG, Lifetech #A11037(1:1,000); AF488 goat anti-rat, Lifetech #A11006 (1:1,000); and Vectashield with DAPI, Vector #H-1200. Instrumentation used was as follows: microscope—Nikon Eclipse Ti-S #TI-FLC-E at 40X/0.75, ∞/0.17 WD 0.72; camera— Zyla scMOS, Andor #DG-152VC1E-FI; acquisition software—NIS Elements 64-Bit AR 4.60.00; and data analysis—FIJI.

## In situ hybridization

Sections (4 µm) of lung tumor tissue underwent deparaffinization, followed by treatment with RNAscope hydrogen peroxide for 10 min at RT, and 1× target retrieval reagent at 99°C for 15–30 min. The slides were then treated with RNAscope Protease Plus for 15–30 min at 40°C in the HybEZ Oven. After pretreatment, the slides were treated using the RNAscope 2.5 HD Detection Reagent-BROWN kit per the manufacturer's protocol. After that, the signal was detected for either the negative control probe (dapB), or murine *Cxcl9*. The following reagents were used: RNAScope target retrieval reagents, Advanced Cell Diagnostics (ACD) #322000; RNAScope wash buffer reagents, ACD #310091; RNAScope 2.5 HD Detection Reagent-BROWN, ACD #322310; RNAScope H2O2 & Protease Plus Reagents, ACD #322330; RNAScope Negative Control Probe_dapB, ACD #310043; RNAScope Probe Mm-Cxcl9, ACD #489341; HybEZ II Oven, ACD #321710/321720; Humidity Control Tray, ACD #310012; EZ-Batch Wash Tray, ACD #310019; and EZ-Batch Slide Holder, ACD #310017. . Instrumentation used was as follows: Microscope/camera—Olympus BX41 System at 40×/0.65, ∞0.17/FN22; acquisition software—SPOT; and data analysis— FIJI.

## Immune cell flow cytometry

Mice were euthanized between 2 and 4 wk post-tumor cell injection. Tumor-bearing left lung lobes were excised, mechanically dissociated, and incubated at 37°C for 30 min with collagenase type II (8,480 U/ml; Worthington Biochemical), elastase (7.5 mg/ml; Worthington Biochemical), and soybean trypsin inhibitor (2 mg/ml; Worthington Biochemical). After which, single-cell suspensions were made and filtered through 70-µm cell strainers (BD), subjected to red blood cell lysis using hypotonic buffer (0.15 mM $NH_4Cl$, 10 mM $KHCO_3$, and 0.1 mM $Na_2EDTA$, pH 7.2), and filtered again through 40-µm cell strainers (BD) (Kwak et al, 2018). For the "T Cell Phenotypic Panel," single-cell suspensions were stained for 30 min at room temperature, followed by fixation and permeabilization overnight, and intracellular stains for 2 h at 4°C the following day. For the "T Cell Stimulation Panel," single-cell suspensions were stimulated with Brefeldin A solution, Monensin solution, and a cell stimulation

cocktail (PMA/Ionomycin) for 5 h at 37°C. Afterwards, single-cell suspensions were stained with cell surface stains, fixed and permeabilized overnight, and finally stained with intracellular stains the following morning (as the T Cell Phenotypic Panel). Samples were run at the University of Colorado Cancer Center Flow Cytometry Core using the Gallios system (Beckman Coulter). The gating strategy involved excluding debris and cell doublets by light scatter, as well as dead cells by a cell viability dye. All data were analyzed using Kaluza software (Beckman Coulter). Antibodies and reagents used were as follows: Foxp3 Staining Buffer Set, eBioscience #00-5523-00; Brefeldin A, BioLegend #420601; monensin, BioLegend #420701; Cell Stimulation Cocktail, eBioscience #00-4970-93; Anti-mouse PD-1-PE, eBioscience #12-9981-81 (1:200); Anti-Mouse CD69-PECy7, eBioscience #25-0691-81 (1:200); Anti-Mouse CD45-AF700, eBioscience #56-0451-82 (1:50); Anti-Mouse IA/IE Dazzle 594, BioLegend #107648 (1:250); Anti-Mouse CD3e-PerCP Cy5.5, eBioscience #45-0031-82 (1:200); Anti-Mouse CD4-EF450, eBioscience #48-0042-82 (1:200); Anti-Mouse CD8a-APC EF780, eBioscience #47-0081-82 (1:200); Murine Fc Block, eBioscience #14-0161-86 (1:100); V500 rat anti-mouse CD4, BD #560782; Aqua Viability Dye, Thermo Fisher Scientific #L34957 (1:200); Rat Anti-Mouse Isotype for IFNγ-AF488, eBioscience #53-4301-80 (1:80); Anti-Mouse IFNγ-AF488, eBioscience #53-7311-82(1:80); Rat Anti-Mouse Isotype for PerCP Cy5.5-TNFα, BD #560537(1:80); Anti-Mouse PerCP Cy5.5-TNFα, BD #560659(1:80); and VersaComp Antibody Capture Bead Kit, Beckman Coulter #B22804. For the LLC-NT replicates, three left tumor-bearing lung lobes were isolated (three mice used per replicate) and combined to make a single-cell suspension, for a total of nine mice used for three experimental replicates. For the LLC-sh21 replicates, three left tumor-bearing lung lobes were isolated (three mice used per replicate) and combined to make a single-cell suspension, for a total of nine mice used for the three experimental replicates.

## CyTOF analysis

Single-cell suspensions prepared as above were treated with benzonase nuclease (#E1014, 1:10,000; Sigma-Aldrich), stained with cisplatin, and fixed for sample barcoding (Fluidigm). Samples were then combined into one tube, followed by incubation with an Fc receptor–blocking antibody, primary surface antibodies, and secondary surface staining. The cells were then fixed and permeabilized overnight, followed by intracellular stains the next day. After staining, the cells were suspended in Intercalator (Kimball et al, 2018). Single-cell suspensions were run on the Helios mass cytometer as part of the University of Colorado Cancer Center Flow Cytometry Core. Antibodies and reagents used were as follows: 89Y-CD45, Fluidigm, Clone 30-F11; 141Pr-Gr1 (Ly6C/Ly6G), Fluidigm, Clone RB6-8C5; 142Nd-CD11c, Fluidigm, Clone N418; 143Nd-GITR, Fluidigm, Clone DTA1; 144Nd-MHC class I, Fluidigm, Clone 28-14-8; 145Nd-SiglecF-PE/anti-PE, BD, Clone E50-2440/Fluidigm, Clone PE001; 146Nd-CD8a, Fluidigm, Clone 53-6.7; 147Nd-p-H2AX[Ser139], Fluidigm, Clone JBW301; 148Nd-CD11b, Fluidigm, Clone M1/70; 149Sm-CD19, Fluidigm, Clone 6D5; 150Nd-CD25, Fluidigm, Clone3C7; 151Eu-CD64, Fluidigm, Clone X54-5/7.1; 152Sm-CD3e, Fluidigm, Clone 145-2C11; 153Eu-PD-L1, Fluidigm, Clone 10F.9G2; 154Sm-CTLA4, Fluidigm, Clone UC10-4B9; 155Gd-IRF4, Fluidigm, Clone 3E4; 156Gd-CD90.2(Thy-1.2),

Fluidigm, Clone 30-H12; 158Gd-FoxP3, Fluidigm, Clone FJK-16s; 159Tb-PD-1, Fluidigm, Clone RMP1-30; 160Gd-CD80/86-FITC/anti-FITC, BD, Clone 16-10A1/BD Clone BL1/Fluidigm, Clone FIT22; 161Dy-INOS, Fluidigm, Clone 4B10; 162Dy-Tim3, Fluidigm, Clone RMT3-23; 163Dy-CXCR3-APC/anti-APC, BioLegend, Clone CXCR3-173/Fluidigm, Clone APC003; 164Dy-IkBa, Fluidigm, Clone L35A5;165Ho-Beta-catenin (active), Fluidigm, Clone D13A1; 166Er-Arg1, Fluidigm, Clone 6D5; 167Er-NKp46, Fluidigm, Clone 9A1.4; 168Er-Ki-67; Fluidigm, Clone Ki-67; 169Tm-Ly-6A/E (Sca-1), Fluidigm, Clone D7; 170Er-CD103-Biotin/anti-Biotin, BioLegend, Clone 2E7/Fluidigm, Clone 1D4-C5; 171Yb-CD44, Fluidigm, Clone IM7; 172Yb-CD4, Fluidigm, Clone RM4-5; 173Yb-CD117 (ckit), Fluidigm, Clone 2B8; 174Yb-Lag3, Fluidigm, Clone M5/114.15.2;175Lu-CD127, Fluidigm, Clone A7R34; 176Yb-ICOS, Fluidigm, Clone 7E.17G9; 191Ir, 193Ir-Intercalator, Cell-ID; 195Pt Cisplatin 5 μM, Cell-ID; 140Ce, 151Eu, 153Eu, 165Ho, 175Lu Normalization Beads; Cell ID 20-Plex Pd Barcoding Kit, Fluidigm, #201060 (102, 104, 105, 106, 108, and 110 Pd bar codes); and Benzonase, Sigma-Aldrich #E1014-5KU (1:10,000) in HBSS. For naive replicates, left and right lung lobes were combined from one mouse per replicate and made into a single-cell suspension. Three total mice were used for the naive experimental condition. For the LLC-NT replicates, three left tumor-bearing lung lobes were isolated (three mice used per replicate) and combined to make a single-cell suspension, for a total of nine mice used for three biological replicates. For the LLC-sh21 replicates, three left tumor-bearing lung lobes were isolated (three mice used per replicate) and combined to make a single-cell suspension, for a total of nine mice used for three biological replicates.

## PhenoGraph analysis methods

Software for data analysis included R studio (Version 1.0.136), downloaded from the official R Web site (https://www.r-project.org/); the cytofkit package (Release 3.6), downloaded from Bioconductor (https://www.bioconductor.org/packages/3.6/bioc/html/cytofkit.html https://www.bioconductor.org/packages/release/bioc/html/cytofkit.html); Excel 15.13. 14, FlowJo 10.2, GraphPad Prism 7, and Adobe Illustrator CC 2017. The samples were normalized using NormalizerR2013b_MacOSX, downloaded from the Nolan laboratory GitHub page (https://github.com/nolanlab). The normalized files were then debarcoded using SingleCellDebarcoderR2013b_MacOSX, downloaded from the Nolan laboratory GitHub page (https://github.com/nolanlab). Debarcoded and normalized data were subjected to traditional Boolean gating in FlowJo, identifying viable singlet events (191Ir+, 193Ir+, 195Pt–). These events were exported for downstream analysis. All viable singlet (19Ir+, 193Ir+, 195Pt+) events were imported into cytofkit analysis pipeline, and 39 markers were selected for clustering. The merge method "min" was selected (12,255 events from each file used for clustering) and the files were transformed via the cytofAsinh method. Then files were clustered with the PhenoGraph algorithm and tSNE was selected as the visualization method. PhenoGraph identified 35 unique clusters. These results were visualized via the R package "Shiny" where labels, dot size, and cluster color were customized according to cluster identity or phenotype. Plots were examined for expression of various cellular markers (parameters). The algorithm produced multiple .csv files, the files "cluster median data" and "cluster cell percentage" which were used to determine cluster frequency and phenotype.

## RNA-Seq analysis of cancer cells recovered from GFP-transgenic mice

GFP-expressing transgenic mice were implanted with $10^5$ cells as described above. After 2–3 wk of tumor growth, single-cell suspensions of tumor-bearing lung lobes were prepared containing a mixture of GFP-negative cancer cells and GFP-positive host cells. GFP-negative cancer cells were sorted using the MoFlo XDP cell sorter with a 100-$\mu$m nozzle (Beckman Coulter) as part of the University of Colorado Cancer Center Flow Cytometry Shared Resource. The sorting strategy excluded dead cells (via DAPI staining) and cell doublets by light scatter. Total RNA was isolated via the RNeasy Plus Mini kit (QIAGEN). CMT167 and LLC or LLC-NT and LLC-sh21 cells were recovered from 3 to 5 pools of mice consisting of at least four GFP-expressing mice per single pool. Each pool represents an experimental replicate. Total RNA was also isolated from cells in culture at the time of injection. Preparation of the RNA-Seq library was done at the University of Colorado Cancer Center Genomics and Microarray Shared Resource. RNA libraries were constructed using an Illumina TruSEQ stranded mRNA Sample Prep Kit and sequencing was performed using an Illumina HiSEQ 4000 System. Reads from RNA-Seq were processed and aligned to a mouse reference genome (University of California Santa Cruz Mus musculus reference genome build mm[10]) via the TopHat v2 software (Poczobutt et al, 2016a). The aligned read files were then processed by Cufflinks v2.0.2 software to determine the relative abundance of mRNA transcripts (Poczobutt et al, 2016a). Reads are portrayed as fragments per kilobase of exon per million fragments mapped (FPKM). For pathway analysis of final FPKM files, various analysis platforms, including KEGG and DAVID were used to determine the most highly enriched pathways between experimental conditions. However, the importance of the IFNγ response pathway was only determined through gene set enrichment analysis of LLC and CMT167 experimental conditions. RNA-Seq data generated from the LLC and CMT167 cell lines were deposited in the Gene Expression Omnibus repository in 2017. Gene Expression Omnibus accession number: GSE100412. RNA-Seq data generated from the LLC-NT and LLC-sh21 cell lines were deposited in the Gene Expression Omnibus repository in 2019. Gene Expression Omnibus accession number: GSE131271.

### Mutational analysis of RNA-Seq data

LLC or CMT167 RNA-Seq sample data files were run through "Module 1: Variants Detection" of the IMPACT pipeline to determine nonsynonymous mutational burden (http://tanlab.ucdenver.edu/IMPACT/pipeline.html).

### Statistical analysis

Statistical Analyses were performed using the GraphPad Prism 7/8 software. Data are presented as mean ± SEM. A *one- or two-way ANOVA* was used to compare differences in more than two groups. A *t* test was used to compare differences between two groups in data with a normal distribution. In all circumstances, $P$-values ≤ 0.05 were considered significant (*$P < 0.05$, **$P < 0.01$, ***$P < 0.001$, and ****$P < 0.0001$).

## Supplementary Information

## Acknowledgements

We would like to thank Lynn Heasley and Rebecca Tucker for the helpful discussions. This work was supported by the National Institutes of Health (NIH) (R01 CA162226 and CA236222 to RA Nemenoff), Colorado Lung SPORE P50 CA058187 to HY Li and RA Nemenoff, the United States Department of Veterans Affairs Biomedical Laboratory Research and Development Service (Career Development Award IK2BX001282 to HY Li), and the NIH/National Center for Advancing Translational Sciences (NCATS) Colorado CTSA TL1 TR001081 to BL Bullock. The University of Colorado Cancer Center Flow Cytometry and the Genomics and Microarray Shared Resources is supported by NIH P30CA046934. The University of Colorado Cancer Center Flow Cytometry Core Facility is funded through a support grant from the National Cancer Institute (P30CA046934). Imaging experiments were performed in the University of Colorado Anschutz Medical Campus Advanced Light Microscopy Core supported in part by NIH/NCATS Colorado Clinical and Translational Sciences Grant Number UL1 TR001082.

### Author Contributions

BL Bullock: conceptualization, data curation, software, formal analysis, funding acquisition, validation, visualization, project administration, and writing—original draft, review, and editing.
AK Kimball: data curation, software, formal analysis, validation, visualization, methodology, and writing—original draft, review, and editing.
JM Poczobutt: data curation, software, formal analysis, validation, methodology, and writing—original draft, review, and editing.
AJ Neuwelt: data curation, formal analysis, validation, methodology, and writing—original draft, review, and editing.
HY Li: conceptualization, resources, supervision, funding acquisition, investigation, methodology, and writing—review and editing.
AM Johnson: data curation, validation, and writing—review and editing.
JW Kwak: data curation, validation, and writing—review and editing.
EK Kleczko: data curation, validation, and writing—review and editing.
RE Kaspar: data curation, validation, and writing—review and editing.
EK Wagner: data curation, validation, and writing—review and editing.
K Hopp: conceptualization, methodology, and writing—review and editing.
EL Schenk: methodology and writing—review and editing.
MCM Weiser-Evans: conceptualization, supervision, investigation, methodology, and writing—review and editing.
ET Clambey: conceptualization, data curation, software, formal analysis, supervision, investigation, methodology, and writing—review and editing.
RA Nemenoff: conceptualization, resources, software, supervision, funding acquisition, investigation, methodology, project administration, and writing—original draft, review, and editing.

### Conflict of Interest Statement

The authors declare that they have no conflict of interest.

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
