## [Reviewer comments · Life Science Alliance]

Life Science Alliance

Tumor-Intrinsic Response to IFN γ Shapes the Tumor Microenvironment and Anti-PD-1 Response in NSCLC

Bonnie Bullock, Abigail Kimball, Joanna Poczobutt, Howard Li, Jeff Kwak, Alexander Neuwelt, Amber Johnson, Emily Kleczko, Rachael Kaspar, Emily Wagner, Katharina Hopp, Erin Schenk, Mary Weiser-Evans, Eric Clambey, and Raphael Nemenoff

DOI: <https://doi.org/10.26508/lsa.201900328>

Corresponding author(s): Raphael Nemenoff, UCD-HSC

Review Timeline:	Submission Date:	2019-01-30
	Editorial Decision:	2019-03-04
	Revision Received:	2019-04-12
	Editorial Decision:	2019-05-06
	Revision Received:	2019-05-13
	Accepted:	2019-05-14

Scientific Editor: Andrea Leibfried

Transaction Report:

March 4, 2019

Re: Life Science Alliance manuscript #LSA-2019-00328-T

Prof. Raphael A Nemenoff
UCD-HSC
Medicine
12700 E. 19th Ave.
Aurora, CO 80045

Dear Dr. Nemenoff,

Thank you for submitting your manuscript entitled "Tumor-Intrinsic Response to IFN γ Shapes the Tumor Microenvironment and Anti-PD-1 Response in NSCLC" to Life Science Alliance. The manuscript was assessed by expert reviewers, whose comments are appended to this letter.

As you will see, the reviewers appreciate your data and provide constructive input on how to further strengthen your manuscript prior to publication. We would thus like to invite you to provide a revised version of your work, addressing the reviewers individual concerns. Elucidating how SOCS1 is silenced and the distinguished features of the lung tumor microenvironment as compared to the flank tumor microenvironment (see also report from reviewer #2) are not mandatory for acceptance here, but all other points need to get satisfactorily addressed.

Thank you for this interesting contribution to Life Science Alliance. We are looking forward to receiving your revised manuscript.

Sincerely,

Andrea Leibfried, PhD
Executive Editor
Life Science Alliance
Meyershofstr. 1
69117 Heidelberg, Germany
t +49 6221 8891 502
e a.leibfried@life-science-alliance.org
www.life-science-alliance.org

B. MANUSCRIPT ORGANIZATION AND FORMATTING:

Reviewer #1 (Comments to the Authors (Required)):

In this manuscript, Bullock et al compared two murine lung cancer cell lines with different responses to anti-PD1 treatment. They identified that IFN γ signaling in cancer cells plays a role in resistance to anti-PD1. Knockout IFN γ receptor renders the sensitive cell line resistant, and inhibition of SOS2,

the inhibitory molecule of IFN γ receptor, did the opposite - making the resistant cells sensitive. Single cell RNA-seq analyses revealed alterations of major immune cell populations including T cells and macrophages. These observations are limited to a pair of cell lines and the generality remains to be established. However, the experiments are performed rigorously and the data are of high quality. The insight is of potential relevance.

My only suggestion to the author is to provide more extended discussions on the controversial roles of IFN γ signaling in immunotherapies. Studies from Andy Minn's group suggested that IFN γ signaling confers resistance to checkpoint blockade (Benci et al., 2016), which would intuitively be contradictory to the authors' finding. This needs to be cited and the thoughts of the authors would be much helpful for readers.

Reviewer #2 (Comments to the Authors (Required)):

In their manuscript titled "Tumor-Intrinsic Response to IFN γ Shapes the Tumor Microenvironment and Anti-PD-1 Response in NSCLC", Bullock and colleagues explore mechanisms underlying checkpoint sensitivity in mouse models of NSCLC. Utilizing two cell lines with differential sensitivity to anti-PD-1, the authors found an IFN response signature correlating to effective anti-PD-1 activity. The authors further probed elements of this signature, and found silencing of *Ifngr1* in the sensitive CMT167 line led to silencing of the pathway and ablated anti-PD-1 efficacy, while silencing an inhibitor of IFN signaling, *Socs1*, had the inverse effect in LLC cells, leading to profound anti-PD-1 effect. The authors further analyzed the immune compartments present in the LLC tumor microenvironment altered by *Socs1* silencing prior to anti-PD-1 therapy, and found an altered myeloid environment and slight alterations in T cell populations. This work advances our mechanistic understanding of key factors influencing checkpoint blockade therapy, and is a timely and interesting work. However, there are several issues that should be addressed to solidify the dataset and make the manuscript more compelling for publication.

Major Concerns:

- Pathway analysis leading to identification of IFN signaling is not shown.
- RNASeq data not appropriate for calling mutations; need whole genome sequencing experiments.
- Demonstration of IFN expression in tumor not shown.
- Expression of *Cxcl9*, *Cxcl10*, *Cd274*, and *Ciita* only shown on the mRNA level in Figures 1 and 2; protein expression (ELISAs, Western blots) needed to confirm their regulation.
- Need at least one in vivo experiment showing main phenotypes with multiple shRNAs, both those targeting *Ifngr1* and targeting *Socs1*, as the concern for off-target effects is very relevant.
- Cohort size in Figure 2G should be expanded- it appears that there may be an effect of *Ifngr1* knockdown itself in the absence of anti-PD-1, but with the limited cohort size it is hard to say.
- How is *SOCS1* silenced in CMT167? Methylation? mRNA instability?
- What are the critical features of the lung TME that are absent in the flank that lead to a lack of anti-PD-1 response in the LLC tumors with *SOCS1* knockdown? This data is presented as a strong indicator of the necessity of the lung TME, and it is compelling, but some (at least cursory) mechanistic exploration of the flank is required.
- The myeloid population data is highly correlative, with no mechanistic exploration. An add back or depletion of one or two candidate populations altering anti-PD-1 efficacy would make the data much more compelling; the global PD-L1 knockout, while sufficient to argue the necessity of host PD-L1, is insufficient to argue for the myeloid populations discussed in depth.
- How do the authors explain the robust differences in CD8 functionality in LLC-NT and LLC-sh21 and no difference in tumor burden prior to anti-PD-1 therapy? Is this difference due to the tumor

itself? Differences in priming?

Minor Concerns:

- Comparisons in Cxcl9, Cxcl10, Cd274, and Ciita expression in Figure 1 should be evaluated between IFN treatments as well as between cell lines.
- Data shown in Figures 2A-E and Figure 3C are a bit obscured by having the parental lines on the graphs as well- compresses axes and makes the most relevant comparisons, between NT and knockdowns, hard to see. Maybe for the main figures have only the NT and knockdown lines, and a new supplemental figure with the complete dataset as shown now?
- Inconsistency in time points of IFN stimulation and evaluation of p-STAT1, STAT1, and SOCS1. Particularly concerning in Supplemental Figure 2, where the time points evaluated are 15 minutes and 48 hours, with nothing in between.
- A bit confused about how 9 lobes become 3 replicates in Figure 4 and Supplemental Figure 3. Are these 3 replicates showing the consistency between replicates from the same mouse, or different mice, or pooled samples? It's a bit hard to believe that a lobe with basically no tumor burden and one with high tumor burden are going to have the exact same immune composition, which is what seems to be argued here.
- It may be best to show the enhanced accumulation of the T cells before showing their enhanced activity.

Reviewer #3 (Comments to the Authors (Required)):

In this manuscript Bullock et al. show that the differential sensitivity to anti-PD-1 blockade between two murine Kras-mutant lung cancer cell lines (LLC and CMT167) can be explained by their response to *lfn-gamma*. This is an excellent, well-executed study that reinforces and builds on data concerning the central importance of IFN- γ in checkpoint blockade. The authors discover that the *Socs1* protein is a node that can be modulated in LLC cells to gain sensitivity to anti-PD-1 and also highlight unresolved complexity in our understanding of checkpoint blockade; the significance of the TME (subcutaneous vs. orthotopic tumor cell site injections) and the relevance of tumor cell-intrinsic PD-L1 levels. Future studies-building on this model--to determine the cell type(s) in which PD-L1 expression is important for anti-PD-1 sensitivity, will be valuable.

The authors' findings can be summarized:

- (i) In cells grown orthotopically, a differential *lfn-gamma* gene signature emerges between CMT167 (anti-PD-1 responsive) and LLC (anti-PD-1 unresponsive) cells.
- (ii) In vitro, CMT167 cells respond to *lfn-gamma* whereas LLC cells have a significantly blunted response.
- (iii) Reduction of *lfngr1* levels in CMT167 cells reverses their sensitivity to PD-1 blockade.
- (iv) Higher basal levels of *Socs1* mRNA and protein in LLC cells can account for the lack of *lfn-gamma* responsiveness in vitro and inhibition of *Socs1* by shRNA is sufficient to sensitize orthotopically injected LLC cells to PD-1 blockade.
- (v) Analysis of the TME (tumor microenvironment) between resistant and sensitive (*Socs1* reduced) LLC cells reveals a modest difference in PD-L1 expression on alveolar macrophages but not on tumor cells.
- (vi) The sensitivity of the LLC-*Socs1*-reduced cell line to anti PD-1 is independent of PD-L1 expression on the tumor cell (shown by orthotopic injection into PD-L1 knockout animals) and may therefore be dependent on its expression in the TME (e.g. alveolar macrophage).

Each of these conclusions has been reached from a reasonable amount of experimental data and analysis. However, minor changes/additions would add to the clarity of the manuscript and are highlighted below as it relates to the appropriate figure/section.

Methods:

-In 'anti-PD-1 Treatment' the dose (units) of anti-PD-1 seems to have been inadvertently removed (200?).

Figure1:

(A) Genes that are routinely assessed by q-PCR (Cxcl9, Cxcl10 and Cd274) in this and later figures are not shown in the heatmap. It would be useful to see the differential expression of these genes in the same context as the IFN-gamma signature genes. The authors mention in the text that 'pathway analysis' was used to discover the differential IFN-gamma signature (and 'KEGG pathway analysis' in the figure legend). It would be helpful to see a pathway enrichment table showing the significance (statistical) of this signature, among others.

(F) The induction of total STAT1 at 1hr and 2hr in LLC cells is surprising. How representative is this finding? If RNA-seq analysis was performed of in vitro lfn-gamma treated LLC and CMT167 cells (24hrs), presentation of this data would significantly strengthen the figure.

Figure2:

(C & D) The y-axis for Cxcl10 and Cd274 is very different from that in Figure 1C and 1D with respect to the induction of these genes in CMT167 cells in response to lfn-gamma. Is the time point or amount of IFN-gamma different? This is also true for Supplemental Figure 2 (B & C).

Figure4:

(A) The population numbers would be easier to follow if presented in bold typeface. A description (in a supplementary table) of the antibody combinations (+/-, high/mid/low?) that resulted in the ~35 populations would be helpful.

(D & E) An exact description of what antibody panel is represented in cluster #2 (vs #1 and #26) and #3 (vs. #4) should be included.

Figure7:

(D & E). The designation of tumor 'edge' and 'middle' is not entirely clear. An adjacent H&E stain or more formal quantitation of this finding would benefit its inclusion.

Supplemental Figure 3.

Is the data in this figure and Figure 4 from the same experiment?

(A) Is this data shown in contrast to Figure 4D (cluster #5) and Supplemental Figure 3D where a reduction in LLC-sh21 tumor cells is documented or a reflection of methodology (calipers vs. flow cytometry)?

Discussion:

In an in vivo CRISPR-Cas9 screen for genes that are depleted or enriched in B16 melanoma cells after immunotherapy, Socs1 was depleted (Manguso et al. Nature 2017; Figure2). A mention and citation of this finding should be included.

Reviewer #1

1) *My only suggestion to the author is to provide more extended discussions on the controversial roles of IFN γ signaling in immunotherapies. Studies from Andy Minn's group suggested that IFN γ signaling confers resistance to checkpoint blockade (Benci et al., 2016), which would intuitively be contradictory to the authors' finding. This needs to be cited and the thoughts of the authors would be much helpful for readers.*

We have provided more in-depth discussion about the contradictory roles of IFN γ in the discussion section. The Benci manuscript provides data showing that chronic IFN γ signaling can lead to resistance to immune checkpoint inhibitors through multiple mechanisms including the induction of other checkpoints. Our model involves a fairly short-term treatment with anti-PD-1, which may account for the lack of resistance. On p.17 of the revised manuscript, we discuss the possibility that longer treatment of our tumors would result in acquired resistance.

Reviewer #2

Major Concerns:

1) *Pathway analysis leading to identification of IFN signaling is not shown.*

We have provided the Gene Set Enrichment Analysis for both the LLC and CMT167 cell lines in a supplementary excel file entitled: **Supplemental Table 1**. These data are discussed on p.6.

2) *RNASeq data not appropriate for calling mutations; need whole genome sequencing experiments.*

While we agree with Reviewer #2 that RNA-Seq is not the most ideal technique to call mutations, there are published resources denoting high specificity and sensitivity using this method: <https://doi.org/10.1016/j.ajhg.2013.08.008>

In order to validate our RNA-Seq mutation calls, we agree that WES or WGS would be necessary. We will use these techniques in the future. We have added **Supplemental Table 2**, discussed on p.6, to show the mutations called via RNA-Seq in the LLC and CMT167 cell lines *in vitro*.

3) *Demonstration of IFN expression in tumor not shown.*

We have included a graph showing IFN γ message via RNA isolated from tumor homogenate from LLC or CMT167 tumors relative to *Gapdh* levels (**Supplemental Figure S1A**), discussed on p.6. This data suggests that both tumors express IFN γ , as well as the capacity for signaling through IFNGR1 and IFNGR2 (**Supplemental Figure S2A-C**).

4) *Expression of Cxcl9, Cxcl10, Cd274, and Ciita only shown on the mRNA level in Figures 1 and 2; protein expression (ELISAs, Western blots) needed to confirm their regulation.*

We have confirmed the protein expression of CXCL9, CXCL10 (by ELISA), PD-L1, MHCII--a standard readout of the transcriptional regulator CIITA's activity, and 2 MHC I genes (H2-D, H2-K) via flow cytometry in parental LLC and CMT167 cells (**Supplemental Figure S1B-H**) \pm IFN γ and \pm Ruxolitinib. We have performed the same experiments in LLC-NT versus LLC-sh21 cells (**Supplemental Figure S4A-F**) since *Socs1* KD LLCs are the focus of the vast majority of the paper. We have also confirmed a decrease in PD-L1 expression (flow cytometry) in the CMT-sh68sc3 cells versus CMT-NT cells after IFN γ treatment (10ng/mL; 72 hours) (**Supplemental Figure S2J**).

5) Need at least one *in vivo* experiment showing main phenotypes with multiple shRNAs, both those targeting *Ifngr1* and targeting *Socs1*, as the concern for off-target effects is very relevant. We have discussed in the manuscript that we cannot completely rule out off target effects based on using only one shRNA *in vivo* in the “Silencing *Ifngr1* in CMT167 Confers Decreased Response to IFN γ and Resistance to Anti-PD-1 Therapy” section on p.8. However, as is common in the field, while we have provided *in vitro* data using two separate shRNAs for both *Ifngr1* and *Socs1*, only one shRNA was selected for *in vivo* studies. An example of a similar approach, characterizing one shRNA *in vivo* has been used by Gao et al (*Cell* 167:397-404), which is referenced in our manuscript. Carrying out all of the *in vivo* studies with a second shRNA would require repeating most of the data in the manuscript. If we were to repeat even one *in vivo* experiment for both the *Ifngr1* and *Socs1* KDs, each shRNA would require 40 mice, as well as anti-PD-1 antibody or control. The shRNA for *in vivo* studies was chosen based on being the better of the two knockdowns functionally. If we were to use the lesser functional shRNA knockdown to repeat *in vivo* experiments, we still might not be able to rule out off-target effects.

Finally, **Supplemental Table 4** shows a list of differentially expressed genes between the LLC-NT and LLC-sh21 tumors *in vivo*. Based on our strict q-value cutoff of $q < 0.05$, there were only 44 differentially expressed genes between the two experimental conditions. Many of these genes are in the interferon response pathway, or related pathways, suggesting limited off-target effects of the shRNA.

6) Cohort size in Figure 2G should be expanded- it appears that there may be an effect of *Ifngr1* knockdown itself in the absence of anti-PD-1, but with the limited cohort size it is hard to say. For your reference, we have provided the following graph whereby CMT-NT or CMT-sh68sc3 were orthotopically injected and grown for 3 weeks without treatment in a separate experiment from previous Figure 2G (now **Figure 3G**). This data indicates that there is not a significant difference in tumor volume at this time point, but we cannot rule out changes in the tumor microenvironment.

7) How is *SOCS1* silenced in CMT167? Methylation? mRNA instability?

We speculate that the mechanism of silencing in the CMT167 cell line is not due to methylation, since these cells can induce *Socs1* mRNA after treatment with IFN γ (**Supplemental Fig S2I, Fig S3B**). However, **Figure 4B** shows that despite treatment with IFN γ , *SOCS1* protein expression is undetectable in CMT167 versus LLC cells. We would therefore propose that posttranscriptional regulation (miRNA, mRNA stability, or translational control) is occurring.

8) What are the critical features of the lung TME that are absent in the flank that lead to a lack of anti-PD-1 response in the LLC tumors with SOCS1 knockdown? This data is presented as a strong indicator of the necessity of the lung TME, and it is compelling, but some (at least cursory) mechanistic exploration of the flank is required.

We have previously shown that the responsiveness of CMT167 tumors to anti-PD-1 therapy is lost in subcutaneous tumors. This was attributed, at least in part to the role of alveolar macrophages, which are only present in orthotopic lung tumors (Li et al, *Cancer Immunol Res* 2017: 5:767-777). In addition, this study reported much higher levels of Treg cells in flank tumors, suggesting that there are distinct immunosuppressive pathways in the two settings.

9) The myeloid population data is highly correlative, with no mechanistic exploration. An add back or depletion of one or two candidate populations altering anti-PD-1 efficacy would make the data much more compelling; the global PD-L1 knockout, while sufficient to argue the necessity of host PD-L1, is insufficient to argue for the myeloid populations discussed in depth. We agree, as discussed in the point above, we are exploring a mechanism to characterize individual myeloid roles in our next paper. For your reference, we have provided the following graphs as a starting point for our next manuscript:

Based on the RNA-Seq of LLC-NT versus LLC-sh21 tumors in **Supplemental Table 4**, we found a significant decrease in tumor-intrinsic *Ccl2* expression in *Socs1* KD LLC-sh21 cells *in vivo*. *Ccl2* is a chemokine that attracts monocytes and macrophages to sites of inflammation via its cognate receptor, CCR2. In previously published data that we cited, we performed RNA-Seq on RNA extracted from specific macrophage subsets in parental LLC tumors (Poczobutt et al *J Immunol* 2016: 2847-59). We noticed that CCR2 expression was highest in the Recruited Macrophage (PD-L1 mid in new **Fig 6A-B, 6D**) and Monocyte populations (PD-L1 lo in new **Fig 6A-C**), and not detectable in resident alveolar macrophages (PD-L1 hi in new **Fig 6A-B, 6E**). Therefore, one reason we believe LLC-sh21 tumors are sensitized to anti-PD-1, is because they recruit less PD-L1 lo monocytes which have the highest expression of CCL2's cognate receptor CCR2. Thus we will begin to explore specific monocyte and macrophage subsets and their recruitment to the tumor microenvironment by altering either tumor-intrinsic *Ccl2* expression or using pharmacologic inhibitors targeting this signaling pathway.

10) How do the authors explain the robust differences in CD8 functionality in LLC-NT and LLC-sh21 and no difference in tumor burden prior to anti-PD-1 therapy? Is this difference due to the tumor itself? Differences in priming?

While the LLC-NT and LLC-sh21 tumors are approximately the same size, this is an insensitive measurement of their composition. We have observed increases in some populations (e.g. T cells) and decreases in others (cancer cells, and PD-L1^{lo} macrophages). There may be differences in T cell priming. Our *in vivo* RNA-Seq analysis (**Supplemental Table 4**) shows that there are three MHC Class I genes with significantly higher mRNA expression in LLC-sh21 cancer cells. This data suggests that *Socs1* KD tumors could be recognized through increased interactions with specifically CD8 T cells, which we have also assessed *in vitro* looking at expression of two MHC Class I genes (H2-D, H2K: **Supplemental Figure S4E-F**).

Minor Concerns:

11) Comparisons in *Cxcl9*, *Cxcl10*, *Cd274*, and *Ciita* expression in Figure 1 should be evaluated between IFN treatments as well as between cell lines.

We have performed the statistical analysis as requested on previous Figure 1-now new **Figure 2**.

12) Data shown in Figures 2A-E and Figure 3C are a bit obscured by having the parental lines on the graphs as well- compresses axes and makes the most relevant comparisons, between NT and knockdowns, hard to see. Maybe for the main figures have only the NT and knockdown lines, and a new supplemental figure with the complete dataset as shown now?

We have removed the parental cell line and the positive or negative control for the main figures (**Figures 3 and 4**), while showing the complete datasets in supplemental figures (**Supplemental Figures S2 and S3**).

13) Inconsistency in time points of IFN stimulation and evaluation of p-STAT1, STAT1, and SOCS1. Particularly concerning in Supplemental Figure 2, where the time points evaluated are 15 minutes and 48 hours, with nothing in between.

Supplemental Figure 2 (now **Supplemental Fig S3C**) is just shown to reiterate that signaling in the LLC-sh21 cells is altered at 15 minutes, and is sustained (48 hours). We have performed other immunoblots that are not included in this manuscript at time points between. We found similar findings to the data provided.

14) A bit confused about how 9 lobes become 3 replicates in Figure 4 and Supplemental Figure 3. Are these 3 replicates showing the consistency between replicates from the same mouse, or different mice, or pooled samples? It's a bit hard to believe that a lobe with basically no tumor burden and one with high tumor burden are going to have the exact same immune composition, which is what seems to be argued here.

In the one CyTOF experiment, a naïve mouse was used as a control and was compared to LLC-NT and LLC-sh21 tumors. This experiment involved injecting mice on the same day with tumor cells, as well as sacrificing mice on the same day. For the naïve samples, each replicate consists of lungs from 1 mouse. Namely, both the left (1 lobe) and right lung lobes (4 lobes) were isolated and made into a single cell suspension. Thus, there are 3 total mice used for the naïve experimental condition. For the LLC-NT samples, each replicate consists of 3 left tumor-bearing lung lobes. These 3 lung lobes were combined (pooled) into a single cell suspension per replicate. Thus, there are 9 total mice used for the LLC-NT experimental condition. For the LLC-sh21 samples, it is the same as the LLC-NT experimental condition, whereby 9 left lung lobes total were used and each replicate had 3 lung lobes pooled. Therefore, new **Figure 5A** represents all 3 replicates combined, while new **Supplemental Fig S5C** shows what each

replicate (1 mouse per naïve replicate; 3 mice per tumor replicates) looks like. Since the tumor replicates are a pool of 3 mice, we note that each replicate is similar enough to get statistical significance of several clusters.

15) It may be best to show the enhanced accumulation of the T cells before showing their enhanced activity.

This is the order in the revised manuscript with representative images and a quantification of CD3 and CD8 T Cells per high power field (immunofluorescence), followed by CD8 T cell activation (flow cytometry).

Reviewer #3

1) Methods: In 'anti-PD-1 Treatment' the dose (units) of anti-PD-1 seems to have been inadvertently removed (200?).

We have corrected this (200 μ g).

*2) Figure1: (A) Genes that are routinely assessed by q-PCR (Cxcl9, Cxcl10 and Cd274) in this and later figures are not shown in the heatmap. It would be useful to see the differential expression of these genes in the same context as the IFN-gamma signature genes. The authors mention in the text that 'pathway analysis' was used to discover the differential IFN-gamma signature (and 'KEGG pathway analysis' in the figure legend). It would be helpful to see a pathway enrichment table showing the significance (statistical) of this signature, among others. As was noted by Reviewer #2, we have included this information in new **Figure 1**, and **Supplemental Table 1**.*

3) (F) The induction of total STAT1 at 1hr and 2hr in LLC cells is surprising. How representative is this finding? If RNA-seq analysis was performed of in vitro lfn-gamma treated LLC and CMT167 cells (24hrs), presentation of this data would significantly strengthen the figure. Upon repeating these blots with old samples as well as new replicates of the parental LLC line and the LLC-NT line, we noticed that STAT1 induction is not consistently detectable until about 4 hours after IFN γ treatment, although there is some variability at earlier time points. We will consider RNA-Seq analysis on LLC and CMT167 cells \pm IFN γ in the future.

4) Figure2: (C & D) The y-axis for Cxcl10 and Cd274 is very different from that in Figure 1C and 1D with respect to the induction of these genes in CMT167 cells in response to lfn-gamma. Is the time point or amount of IFN-gamma different? This is also true for Supplemental Figure 2 (B & C).

The original experiments between parental CMTs and LLCs were done with 100ng/mL IFN γ . One potential reason the axes are different between these figures involves the amount of IFN γ used to treat these cells (100ng/mL vs. 10ng/mL in subsequent experiments). Another reason is that the standard curves used different cDNA samples in the original experiments and thus should have slightly different axes depending on cDNA dilution and sample types in the standard mix. For all experiments looking at mRNA expression via qRT-PCR, cells were treated for 24 hours regardless of IFN γ concentration.

5) Figure4: (A) The population numbers would be easier to follow if presented in bold typeface. A description (in a supplementary table) of the antibody combinations (+/-, high/mid/low?) that resulted in the ~35 populations would be helpful.

We have provided **Supplemental Table 3** that shows markers each cluster is positive for, as well as a breakdown of the requested information. We made the numbers in new **Figure 5A**

clearer. New **Supplemental Figures S5B** and **S6A** show a visual representation of **Supplemental Table 3** by marker.

6) (D & E) An exact description of what antibody panel is represented in cluster #2 (vs #1 and #26) and #3 (vs. #4) should be included.

As stated in point #5, we have provided **Supplemental Table 3** to denote specifically how these clusters differ by marker expression.

7) *Figure 7: (D & E). The designation of tumor 'edge' and 'middle' is not entirely clear. An adjacent H&E stain or more formal quantitation of this finding would benefit its inclusion.* We have provided an example of a tumor positive for *Cxcl9* and its adjacent H&E stain in new **Figure 8B** to clarify the designation of tumor edge versus middle. We have tried to quantify positive *Cxcl9* staining using the FIJI plugin, "Trainable Weka Segmentation." Unfortunately, the program was unable to distinguish positive staining, though we are currently looking for other means of quantification in future RNAScope (ISH) experiments.

8) *Supplemental Figure 3. Is the data in this figure and Figure 4 from the same experiment?* Yes, this is the same experiment.

(A) *Is this data shown in contrast to Figure 4D (cluster #5) and Supplemental Figure 3D where a reduction in LLC-sh21 tumor cells is documented or a reflection of methodology (calipers vs. flow cytometry)?*

For the CyTOF experiment, we used tumors that were of similar size (new **Supplemental Figure S5A**), as measured by digital calipers. Despite the tumors being of similar size, there were less cancer cells by percentage in LLC-sh21 tumor bearing lungs, and more immune cells at this early time point versus LLC-NT tumor bearing lungs. We have seen similar results in our RNA-Seq experiments at the terminal time point shown in **Fig 8A, Supplemental Table 4**, and the **SourceDataForFigure8A.xls** file. In LLC-sh21 tumors there was a decreased percentage of live cancer cells, and an increase in live host cells as assessed by the GFP-negative (cancer cell) vs. GFP-positive (host cell) populations before cell sorting for RNA isolation/RNA-Seq. While tumors may be of similar size, they have altered cellular composition. We have included flow cytometry data for your reference at the terminal time point (before cells were sorted for RNA-Seq).

9) Discussion:

In an in vivo CRISPR-Cas9 screen for genes that are depleted or enriched in B16 melanoma cells after immunotherapy, Socs1 was depleted (Manguso et al. Nature 2017; Figure2). A mention and citation of this finding should be included.

We have added this to the Discussion section on p.15 and have added this citation.

May 6, 2019

RE: Life Science Alliance Manuscript #LSA-2019-00328-TR

Prof. Raphael A Nemenoff
UCD-HSC
Medicine
12700 E. 19th Ave.
Aurora, CO 80045

Dear Dr. Nemenoff,

Thank you for submitting your revised manuscript entitled "Tumor-Intrinsic Response to IFN γ Shapes the Tumor Microenvironment and Anti-PD-1 Response in NSCLC". We asked original reviewer #2 and #3 to re-evaluate the revised work. As you will see, reviewer #3 supports publication, in line with the support provided by previous reviewer #1. Reviewer #2, however, still thinks that there is a lack of in-depth exploration of the myeloid populations and of the mechanism of SOCS1 silencing, and the reviewer also still thinks that the in vivo work needs validation with a second shRNA to exclude potential off-target effects. While providing the requested insight and validation would further strengthen your work, our view remains that the mechanism of SOCS1 silencing does not need to get elucidated for publication here. We also think that the other two remaining concerns can get addressed by acknowledging them in the manuscript text. We would thus be happy to publish your paper in Life Science Alliance pending final text revisions to address reviewer #2's concerns as well as revisions necessary to meet our formatting guidelines:

- please deposit the RNA-seq data in an appropriate database and add the accession number in the methods section
- please indicate the number of biological and technical replicates for the q-RT-PCRs performed

A. FINAL FILES:

-- High-resolution figure, supplementary figure and video files uploaded as individual files: See our

detailed guidelines for preparing your production-ready images, <http://www.life-science-alliance.org/authors>

B. MANUSCRIPT ORGANIZATION AND FORMATTING:

Sincerely,

Andrea Leibfried, PhD
Executive Editor
Life Science Alliance
Meyershofstr. 1
69117 Heidelberg, Germany
t +49 6221 8891 502
e a.leibfried@life-science-alliance.org

Reviewer #2 (Comments to the Authors (Required)):

The authors have only addressed some of my concerns. Therefore, I am willing to accept the details the authors provide from previous explorations of lung vs flank differences, however, there is a lack of in-depth exploration of the myeloid populations. The mechanism of SOCS1 silencing remains unexplored and the authors have considered this beyond the scope of the manuscript. The authors should provide a more explicit in-depth discussion of the lack of tumor burden phenotype in sh21 compared to shNT prior to anti-PD-1 therapy while there exists a robust difference in CD8 T cell functionality, as the current discussion is superficial and lacking experimental support to correspond with the observed phenotypes.

Given that shRNA often are associated with offtargeting, concern still remains on the use of a single shRNA for in vivo experiments. I understand that mouse experiments are expensive and time-consuming, but I do consider this to be critical.

Reviewer #3 (Comments to the Authors (Required)):

The authors have adequately addressed the points initially raised and in my opinion, this manuscript now warrants publication.

May 14, 2019

RE: Life Science Alliance Manuscript #LSA-2019-00328-TRR

Prof. Raphael A Nemenoff
UCD-HSC
Medicine
12700 E. 19th Ave.
Aurora, CO 80045

Dear Dr. Nemenoff,

Thank you for submitting your Research Article entitled "Tumor-Intrinsic Response to IFN γ Shapes the Tumor Microenvironment and Anti-PD-1 Response in NSCLC". I appreciate the introduced changes and it is a pleasure to let you know that your manuscript is now accepted for publication in Life Science Alliance. Congratulations on this interesting work.

DISTRIBUTION OF MATERIALS:

Again, congratulations on a very nice paper. I hope you found the review process to be constructive and are pleased with how the manuscript was handled editorially. We look forward to future exciting submissions from your lab.

Sincerely,
